# Orthogonal NMF through Subspace Exploration

**Megasthenis Asteris**
The University of Texas at Austin
megas@utexas.edu

**Dimitris Papailiopoulos**
University of California, Berkeley
dimitrisp@berkeley.edu

**Alexandros G. Dimakis**
The University of Texas at Austin
dimakis@austin.utexas.edu

## Abstract

Orthogonal Nonnegative Matrix Factorization (ONMF) aims to approximate a nonnegative matrix as the product of two $k$-dimensional nonnegative factors, one of which has orthonormal columns. It yields potentially useful data representations as superposition of disjoint parts, while it has been shown to work well for clustering tasks where traditional methods underperform. Existing algorithms rely mostly on heuristics, which despite their good empirical performance, lack provable performance guarantees.

We present a new ONMF algorithm with provable approximation guarantees. For any constant dimension $k$, we obtain an additive EPTAS without any assumptions on the input. Our algorithm relies on a novel approximation to the related Nonnegative Principal Component Analysis (NNPCA) problem; given an arbitrary data matrix, NNPCA seeks $k$ nonnegative components that jointly capture most of the variance. Our NNPCA algorithm is of independent interest and generalizes previous work that could only obtain guarantees for a single component.

We evaluate our algorithms on several real and synthetic datasets and show that their performance matches or outperforms the state of the art.

## 1 Introduction

**Orthogonal NMF** The success of Nonnegative Matrix Factorization (NMF) in a range of disciplines spanning data mining, chemometrics, signal processing and more, has driven an extensive practical and theoretical study [1, 2, 3, 4, 5, 6, 7, 8]. Its power lies in its potential to generate meaningful decompositions of data into non-subtractive combinations of a few nonnegative parts.

Orthogonal NMF (ONMF) [9] is a variant of NMF with an additional orthogonality constraint: given a real *nonnegative* $m \times n$ matrix $\mathbf{M}$ and a target dimension $k$, typically much smaller than $m$ and $n$, we seek to approximate $\mathbf{M}$ by the product of an $m \times k$ nonnegative matrix $\mathbf{W}$ with orthogonal (w.l.o.g, orthonormal) columns, and an $n \times k$ nonnegative matrix $\mathbf{H}$. In the form of an optimization,

$$\text{(ONMF)} \qquad \mathcal{E}_\star \triangleq \min_{\substack{\mathbf{W} \geq \mathbf{0}, \ \mathbf{H} \geq \mathbf{0} \\ \mathbf{W}^\top \mathbf{W} = \mathbf{I}_k}} \|\mathbf{M} - \mathbf{W}\mathbf{H}^\top\|_{\mathrm{F}}^2. \qquad (1)$$

Since $\mathbf{W}$ is nonnegative, its columns are orthogonal if and only if they have disjoint supports. In turn, each row of $\mathbf{M}$ is approximated by a scaled version of a single (transposed) column of $\mathbf{H}$.

Despite the admittedly limited representational power compared to NMF, ONMF yields sparser part-based representations that are potentially easier to interpret, while it naturally lends itself to certain applications. In a clustering setting, for example, $\mathbf{W}$ serves as a cluster membership matrix and the

columns of $\mathbf{H}$ correspond to $k$ cluster centroids [9, 10, 11]. Empirical evidence shows that ONMF performs remarkably well in certain clustering tasks, such as document classification [6, 11, 12, 13, 14, 15]. In the analysis of textual data where $\mathbf{M}$ is a words by documents matrix, the orthogonal columns of $\mathbf{W}$ can be interpreted as topics defined by disjoint subsets of words. In the case of an image dataset, with each column of $\mathbf{M}$ corresponding to an image evaluated on multiple pixels, each of the orthogonal base vectors highlights a disjoint segment of the image area.

**Nonnegative PCA** For any given factor $\mathbf{W} \geq \mathbf{0}$ with orthonormal columns, the second ONMF factor $\mathbf{H}$ is readily determined: $\mathbf{H} = \mathbf{M}^\top \mathbf{W} \geq \mathbf{0}$. This follows from the fact that $\mathbf{M}$ is by assumption nonnegative. Based on the above, it can be shown that the ONMF problem (1) is equivalent to

$$\text{(NNPCA)} \qquad \mathcal{V}_\star \triangleq \max_{\mathbf{W} \in \mathcal{W}_k} \|\mathbf{M}^\top \mathbf{W}\|_{\mathrm{F}}^2, \tag{2}$$

where

$$\mathcal{W}_k \triangleq \left\{ \mathbf{W} \in \mathbb{R}^{m \times k} : \ \mathbf{W} \geq \mathbf{0}, \ \mathbf{W}^\top \mathbf{W} = \mathbf{I}_k \right\}.$$

For arbitrary —*i.e.*, not necessarily nonnegative— matrices $\mathbf{M}$, the non-convex maximization (2) coincides with the Nonnegative Principal Component Analysis (NNPCA) problem [16]. Similarly to vanilla PCA, NNPCA seeks $k$ orthogonal components that jointly capture most of the variance of the (centered) data in $\mathbf{M}$. The nonzero entries of the extracted components, however, must be positive, which renders the problem NP-hard even in the case of a single component ($k = 1$) [17].

**Our Contributions** We present a novel algorithm for NNPCA. Our algorithm approximates the solution to (2) for any real input matrix and is accompanied with global approximation guarantees. Using the above as a building block, we develop an algorithm to approximately solve the ONMF problem (1) on any *nonnegative* matrix. Our algorithm outputs a solution that strictly satisfies both the nonnegativity and the orthogonality constraints. Our main results are as follows:

**Theorem 1.** (NNPCA) *For any $m \times n$ matrix $\mathbf{M}$, desired number of components $k$, and accuracy parameter $\epsilon \in (0, 1)$, our NNPCA algorithm computes $\overline{\mathbf{W}} \in \mathcal{W}_k$ such that*

$$\left\|\mathbf{M}^\top \overline{\mathbf{W}}\right\|_{\mathrm{F}}^2 \ \geq \ (1 - \epsilon) \cdot \mathcal{V}_\star - k \cdot \sigma_{r+1}^2(\mathbf{M}),$$

*where $\sigma_{r+1}(\mathbf{M})$ is the $(r+1)$th singular value of $\mathbf{M}$, in time $T_{\mathsf{SVD}}(r) + O\big(\big(\frac{1}{\epsilon}\big)^{r \cdot k} \cdot k \cdot m\big)$.*

Here, $T_{\mathsf{SVD}}(r)$ denotes the time required to compute a rank-$r$ approximation $\overline{\mathbf{M}}$ of the input $\mathbf{M}$ using the truncated singular value decomposition (SVD). Our NNPCA algorithm operates on the low-rank matrix $\overline{\mathbf{M}}$. The parameter $r$ controls a natural trade-off; higher values of $r$ lead to tighter guarantees, but impact the running time of our algorithm. Finally, note that despite the exponential dependence in $r$ and $k$, the complexity scales polynomially in the ambient dimension of the input.

If the input matrix $\mathbf{M}$ is nonnegative, as in any instance of the ONMF problem, we can compute an approximate orthogonal nonnegative factorization in two steps: first obtain an orthogonal factor $\mathbf{W}$ by (approximately) solving the NNPCA problem on $\mathbf{M}$, and subsequently set $\mathbf{H} = \overline{\mathbf{M}}^\top \mathbf{W}$.

**Theorem 2.** (ONMF) *For any $m \times n$ nonnegative matrix $\mathbf{M}$, target dimension $k$, and desired accuracy $\epsilon \in (0, 1)$, our ONMF algorithm computes an ONMF pair $\mathbf{W}, \mathbf{H}$, such that*

$$\|\mathbf{M} - \mathbf{W}\mathbf{H}^\top\|_{\mathrm{F}}^2 \ \leq \ \mathcal{E}_\star + \epsilon \cdot \|\mathbf{M}\|_{\mathrm{F}}^2,$$

*in time $T_{\mathsf{SVD}}(\frac{k}{\epsilon}) + O\big(\big(\frac{1}{\epsilon}\big)^{k^2/\epsilon} \cdot k \cdot m\big)$.*

For any constant dimension $k$, Theorem 2 implies an additive EPTAS for the relative ONMF approximation error. This is, to the best our knowledge, the first general ONMF approximation guarantee since we impose no assumptions on $\mathbf{M}$ beyond nonnegativity.

We evaluate our NNPCA and ONMF algorithms on synthetic and real datasets. As we discuss in Section 4, for several cases we show improvements compared to the previous state of the art.

**Related Work** ONMF as a variant of NMF first appeared implicitly in [18]. The formulation in (1) was introduced in [9]. Several algorithms in a subsequent line of work [12, 13, 19, 20, 21, 22] approximately solve variants of that optimization problem. Most rely on modifying approaches for NMF to accommodate the orthogonality constraint; either exploiting the additional structural properties in the objective [13], introducing a penalization term [9], or updating the current estimate

in suitable directions [12], they typically reduce to a multiplicative update rule which attains orthogonality only in a limit sense. In [11], the authors suggest two alternative approaches: an EM algorithm motivated by connections to spherical $k$-means, and an augmented Lagrangian formulation that explicitly enforces orthogonality, but only achieves nonnegativity in the limit. Despite their good performance in practice, existing methods only guarantee local convergence.

A significant body of work [23, 24, 25, 26] has focused on Separable NMF, a variant of NMF partially related to ONMF. Sep. NMF seeks to decompose $\mathbf{M}$ into the product of two nonnegative matrices $\mathbf{W}$ and $\mathbf{H}^\top$ where $\mathbf{W}$ contains a permutation of the $k \times k$ identity matrix. Intuitively, the geometric picture of Sep. NMF should be quite different from that of ONMF: in the former, the rows of $\mathbf{H}^\top$ are the extreme rays of a convex cone enclosing all rows of $\mathbf{M}$, while in the latter they

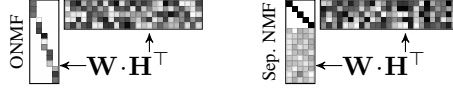

Figure 1: ONMF and Separable NMF, upon appropriate permutation of the rows of $\mathbf{M}$. In the first case, each row of $\mathbf{M}$ is approximated by a single row of $\mathbf{H}^\top$, while in the second, by a nonnegative combination of all $k$ rows of $\mathbf{H}^\top$.

should be scattered in the interior of that cone so that each row of $\mathbf{M}$ has one representative in small angular distance. Algebraically, ONMF factors approximately satisfy the structural requirement of Sep. NMF, but the converse is not true: a Sep. NMF solution is *not* a valid ONMF solution (Fig. 1).

In the NNPCA front, nonnegativity as a constraint on PCA first appeared in [16], which proposed a coordinate-descent scheme on a penalized version of (2) to compute a set of nonnegative components. In [27], the authors developed a framework stemming from Expectation-Maximization (EM) on a generative model of PCA to compute a nonnegative (and optionally sparse) component. In [17], the authors proposed an algorithm based on sampling points from a low-dimensional subspace of the data covariance and projecting them on the nonnegative orthant. [27] and [17] focus on the single-component problem; multiple components can be computed sequentially employing a heuristic deflation step. Our main theoretical result is a generalization of the analysis of [17] for multiple components. Finally, note that despite the connection between the two problems, existing algorithms for ONMF are not suitable for NNPCA as they only operate on nonnegative matrices.

## 2 Algorithms and Guarantees

### 2.1 Overview

We first develop an algorithm to approximately solve the NNPCA problem (2) on any arbitrary — *i.e.*, not necessarily nonnegative— $m \times n$ matrix $\mathbf{M}$. The core idea is to solve the NNPCA problem not directly on $\mathbf{M}$, but a rank-$r$ approximation $\overline{\mathbf{M}}$ instead. Our main technical contribution is a procedure that approximates the solution to the constrained maximization (2) on a rank-$r$ matrix within a multiplicative factor arbitrarily close to 1, in time exponential in $r$, but polynomial in the dimensions of the input. Our Low Rank NNPCA algorithm relies on generating a large number of candidate solutions, one of which provably achieves objective value close to optimal.

The $k$ nonnegative components $\overline{\mathbf{W}} \in \mathcal{W}_k$ returned by our Low Rank NNPCA algorithm on the sketch $\overline{\mathbf{M}}$ are used as a surrogate for the desired components of the original input $\mathbf{M}$. Intuitively, the performance of the extracted nonnegative components depends on how well $\mathbf{M}$ is approximated by the low rank sketch $\overline{\mathbf{M}}$; a higher rank approximation leads to better results. However, the complexity of our low rank solver depends exponentially in the rank of its input. A natural trade-off arises between the quality of the extracted components and the running time of our NNPCA algorithm.

Using our NNPCA algorithm as a building block, we propose a novel algorithm for the ONMF problem (1). In an ONMF instance, we are given an $m \times n$ *nonnegative* matrix $\mathbf{M}$ and a target dimension $k < m, n$, and seek to approximate $\mathbf{M}$ with a product $\mathbf{WH}^\top$ of two nonnegative matrices, where $\mathbf{W}$ additionally has orthonormal columns. Computing such a factorization is equivalent to solving the NNPCA problem on the nonnegative matrix $\mathbf{M}$. (See Appendix A.1 for a formal argument.) Once a nonnegative orthogonal factor $\mathbf{W}$ is obtained, the second ONMF factor is readily determined: $\mathbf{H} = \mathbf{M}^\top \mathbf{W}$ minimizes the Frobenius approximation error in (1) for a given $\mathbf{W}$. Under an appropriate configuration of the accuracy parameters, for any nonnegative $m \times n$ input $\mathbf{M}$ and constant target dimension $k$, our algorithm yields an additive EPTAS for the relative approximation error, without any additional assumptions on the input data.

## 2.2 Main Results

**Low Rank NNPCA**  We develop an algorithm to approximately solve the NNPCA problem on an $m \times n$ real rank-$r$ matrix $\overline{\mathbf{M}}$:

$$\overline{\mathbf{W}}_\star \triangleq \arg\max_{\mathbf{W} \in \mathcal{W}_k} \|\overline{\mathbf{M}}^\top \mathbf{W}\|. \qquad (3)$$

The procedure, which lies in the core of our subsequent developments, is encoded in Alg. 1. We describe it in detail in Section 3. The key observation is that irrespectively of the dimensions of the input, the maximization in (3) can be reduced to $k \cdot r$ unknowns. The algorithm generates a large number of $k$-tuples of $r$-dimensional points; the collec-

---

**Algorithm 1** `LowRankNNPCA`

**input** real $m \times n$ rank-$r$ matrix $\overline{\mathbf{M}}$, $k$, $\epsilon \in (0,1)$
**output** $\mathbf{W} \in \mathcal{W}_k \subset \mathbb{R}^{m \times k}$ {See Lemma 1}
1: $\mathcal{C} \leftarrow \{\}$ {Candidate solutions}
2: $[\overline{\mathbf{U}}, \overline{\mathbf{\Sigma}}, \overline{\mathbf{V}}] \leftarrow \mathrm{SVD}(\overline{\mathbf{M}}, r)$ {Trunc. SVD}
3: **for each** $\mathbf{C} \in \mathcal{N}_{\epsilon/2}^{\otimes k}\left(\mathbb{S}_2^{r-1}\right)$ **do**
4:     $\mathbf{A} \leftarrow \overline{\mathbf{U}}\overline{\mathbf{\Sigma}}\mathbf{C}$ {$\mathbf{A} \in \mathbb{R}^{m \times k}$}
5:     $\widehat{\mathbf{W}} \leftarrow \mathrm{LocalOptW}(\mathbf{A})$ {Alg. 3}
6:     $\mathcal{C} \leftarrow \mathcal{C} \cup \{\widehat{\mathbf{W}}\}$
7: **end for**
8: $\overline{\mathbf{W}} \leftarrow \arg\max_{\mathbf{W} \in \mathcal{C}} \|\overline{\mathbf{M}}^\top \mathbf{W}\|_{\mathrm{F}}^2$

---

tion of tuples is denoted by $\mathcal{N}_{\epsilon/2}^{\otimes k}\left(\mathbb{S}_2^{r-1}\right)$, the $k$th Cartesian power of an $\epsilon/2$-net of the $r$-dimensional unit sphere. Using these points, we effectively sample the column-space of the input $\overline{\mathbf{M}}$. Each tuple yields a feasible solution $\mathbf{W} \in \mathcal{W}_k$ through a computationally efficient subroutine (Alg. 3). The best among those candidate solutions is provably close to the optimal $\overline{\mathbf{W}}_\star$ with respect to the objective in (2). The approximation guarantees are formally established in the following lemma.

**Lemma 1.** *For any real $m \times n$ matrix $\overline{\mathbf{M}}$ with rank $r$, desired number of components $k$, and accuracy parameter $\epsilon \in (0,1)$, Algorithm 1 outputs $\overline{\mathbf{W}} \in \mathcal{W}_k$ such that*

$$\|\overline{\mathbf{M}}^\top \overline{\mathbf{W}}\|_{\mathrm{F}}^2 \geq (1 - \epsilon) \cdot \|\overline{\mathbf{M}}^\top \overline{\mathbf{W}}_\star\|_{\mathrm{F}}^2,$$

*where $\overline{\mathbf{W}}_\star$ is the optimal solution defined in (3), in time $T_{SVD}(r) + O\left(\left(\frac{2}{\epsilon}\right)^{r \cdot k} \cdot k \cdot m\right)$.*

*Proof.* (See Appendix A.2.) □

**Nonnegative PCA**  Given an arbitrary real $m \times n$ matrix $\mathbf{M}$, we can generate a rank-$r$ sketch $\overline{\mathbf{M}}$ and solve the low rank NNPCA problem on $\overline{\mathbf{M}}$ using Algorithm 1. The output $\overline{\mathbf{W}} \in \mathcal{W}_k$ of the low rank problem can be used as a surrogate for the desired components of the original input $\mathbf{M}$. For simplicity, here we consider the case where $\overline{\mathbf{M}}$ is the rank-$r$ approximation of $\mathbf{M}$ obtained by the truncated SVD. Intuitively, the performance of the extracted components on the original data matrix $\mathbf{M}$ will depend on how well the latter is approximated by $\overline{\mathbf{M}}$, and in turn by the spectral decay of the input data. For example, if $\mathbf{M}$ exhibits a sharp spectral decay, which is frequently the case in real data, a moderate value of $r$ suffices to obtain a good approximation. This leads to our first main theorem which formally establishes the guarantees of our NNPCA algorithm.

**Theorem 1.** *For any real $m \times n$ matrix $\mathbf{M}$, let $\overline{\mathbf{M}}$ be its best rank-$r$ approximation. Algorithm 1 with input $\overline{\mathbf{M}}$, and parameters $k$ and $\epsilon \in (0,1)$, outputs $\overline{\mathbf{W}} \in \mathcal{W}_k$ such that*

$$\left\|\mathbf{M}^\top \overline{\mathbf{W}}\right\|_{\mathrm{F}}^2 \geq (1 - \epsilon) \cdot \left\|\mathbf{M}^\top \mathbf{W}_\star\right\|_{\mathrm{F}}^2 - k \cdot \left\|\mathbf{M} - \overline{\mathbf{M}}\right\|_2^2,$$

*where $\mathbf{W}_\star \triangleq \arg\max_{\mathbf{W} \in \mathcal{W}_k} \left\|\mathbf{M}^\top \mathbf{W}\right\|_{\mathrm{F}}^2$, in time $T_{SVD}(r) + O\left(\left(\frac{1}{\epsilon}\right)^{r \cdot k} \cdot k \cdot m\right)$.*

*Proof.* The proof follows from Lemma 1. It is formally provided in Appendix A.3. □

Theorem 1 establishes a trade-off between the computational complexity of the proposed NNPCA approach and the tightness of the approximation guarantees; higher values of $r$ imply smaller $\|\mathbf{M} - \overline{\mathbf{M}}\|_2^2$ and in turn a tighter bound (assuming that the singular values of $\mathbf{M}$ decay), but have an exponential impact on the running time. Despite the exponential dependence on $r$ and $k$, our approach is polynomial in the dimensions of the input $\mathbf{M}$, dominated by the truncated SVD.

In practice, Algorithm 1 can be terminated early returning the best computed result at the time of termination, sacrificing the theoretical approximation guarantees. In Section 4 we empirically evaluate our algorithm on real datasets and demonstrate that even for small values of $r$, our NNPCA algorithms significantly outperforms existing approaches.

**Orthogonal NMF** The NNPCA algorithm straightforwardly yields an algorithm for the ONMF problem (1). In an ONMF instance, the input matrix $\mathbf{M}$ is by assumption nonnegative. Given any $m \times k$ orthogonal nonnegative factor $\mathbf{W}$, the optimal choice for the second factor is $\mathbf{H} = \mathbf{M}^\top \mathbf{W}$. Hence, it suffices to determine $\mathbf{W}$, which can be obtained by solving the NNPCA problem on $\mathbf{M}$.

The proposed ONMF algorithm is outlined in Alg. 2. Given a nonnegative $m \times n$ matrix $\mathbf{M}$, we first obtain a rank-$r$ approximation $\overline{\mathbf{M}}$ via the truncated SVD, where $r$ is an accuracy parameter. Using Alg. 1 on $\overline{\mathbf{M}}$, we compute an orthogonal nonnegative factor $\overline{\mathbf{W}} \in \mathcal{W}_k$ that approximately maximizes (3) within a desired accuracy. The second ONMF factor $\overline{\mathbf{H}}$ is readily determined as described earlier.

---
**Algorithm 2** ONMFS

---
**input** : $m \times n$ real $\mathbf{M} \geq \mathbf{0}$, $r$, $k$, $\epsilon \in (0,1)$
1: $\overline{\mathbf{M}} \leftarrow \text{SVD}(\mathbf{M}, r)$
2: $\overline{\mathbf{W}} \leftarrow \text{LowRankNNPCA}(\overline{\mathbf{M}}, k, \epsilon)$      {Alg. 1}
3: $\overline{\mathbf{H}} \leftarrow \mathbf{M}^\top \overline{\mathbf{W}}$
**output** $\overline{\mathbf{W}}, \overline{\mathbf{H}}$

---

The accuracy parameter $r$ once again controls a trade-off between the quality of the ONMF factors and the complexity of the algorithm. We note, however, that for any target dimension $k$ and desired accuracy parameter $\epsilon$, setting $r = \lceil k/\epsilon \rceil$ suffices to achieve an additive $\epsilon$ error on the relative approximation error of the ONMF problem. More formally,

**Theorem 2.** *For any $m \times n$ real nonnegative matrix $\mathbf{M}$, target dimension $k$, and desired accuracy $\epsilon \in (0,1)$, Algorithm 2 with parameter $r = \lceil k/\epsilon \rceil$ outputs an ONMF pair $\mathbf{W}, \mathbf{H}$, such that*

$$\|\mathbf{M} - \mathbf{W}\mathbf{H}^\top\|_\text{F}^2 \;\leq\; \mathcal{E}_\star + \varepsilon \cdot \|\mathbf{M}\|_\text{F}^2,$$

*in time $T_{\text{SVD}}(\frac{k}{\epsilon}) + O\big((\frac{1}{\epsilon})^{k^2/\epsilon} \cdot (k \cdot m)\big)$.*

*Proof.* (See Appendix A.4.) $\qquad\qquad\qquad\qquad\qquad\qquad\qquad\qquad\qquad\qquad\square$

Theorem 2 implies an additive EPTAS[1] for the relative approximation error in the ONMF problem for any *constant* target dimension $k$; Algorithm 2 runs in time polynomial in the dimensions of the input $\mathbf{M}$. Finally, note that it did not require any assumption on $\mathbf{M}$ beyond nonnegativity.

## 3 The Low Rank NNPCA Algorithm

In this section, we re-visit Alg. 1, which plays a central role in our developments, as it is the key piece of our NNPCA and in turn our ONMF algorithm. Alg. 1 approximately solves the NNPCA problem (3) on a rank-$r$, $m \times n$ matrix $\overline{\mathbf{M}}$. It operates by producing a large, but tractable number of candidate solutions $\mathbf{W} \in \mathcal{W}_k$, and returns the one that maximizes the objective value in (2). In the sequel, we provide a brief description of the ideas behind the algorithm.

We are interested in approximately solving the low rank NNPCA problem (3). Let $\overline{\mathbf{M}} = \overline{\mathbf{U}}\,\overline{\boldsymbol{\Sigma}}\,\overline{\mathbf{V}}^\top$ denote the truncated SVD of $\overline{\mathbf{M}}$. For any $\mathbf{W} \in \mathbb{R}^{m \times k}$,

$$\|\overline{\mathbf{M}}^\top \mathbf{W}\|_\text{F}^2 = \|\overline{\boldsymbol{\Sigma}}\,\overline{\mathbf{U}}^\top \mathbf{W}\|_\text{F}^2 = \sum_{j=1}^k \|\overline{\boldsymbol{\Sigma}}\,\overline{\mathbf{U}}^\top \mathbf{w}_j\|_2^2 = \sum_{j=1}^k \max_{\mathbf{c}_j \in \mathbb{S}_2^{r-1}} \big\langle \mathbf{w}_j,\, \overline{\mathbf{U}}\,\overline{\boldsymbol{\Sigma}}\mathbf{c}_j \big\rangle^2, \qquad (4)$$

where $\mathbb{S}_2^{r-1}$ denotes the $r$-dimensional $\ell_2$-unit sphere. Let $\mathbf{C}$ denote the $r \times k$ variable formed by stacking the unit-norm vectors $\mathbf{c}_j$, $j = 1, \ldots, k$. The key observation is that for a given $\mathbf{C}$, we can efficiently compute a $\mathbf{W} \in \mathcal{W}_k$ that maximizes the right-hand side of (4). The procedure for that task is outlined in Alg. 3. Hence, the NNPCA problem (3) is reduced to determining the optimal value of the low-dimensional variable $\mathbf{C}$. But, first let us we provide a brief description of Alg. 3.

For a fixed $r \times k$ matrix $\mathbf{C}$, Algorithm 3 computes

$$\widehat{\mathbf{W}} \triangleq \arg\max_{\mathbf{W} \in \mathcal{W}_k} \sum_{j=1}^{k} \left\langle \mathbf{w}_j, \, \mathbf{a}_j \right\rangle^2, \qquad (5)$$

where $\mathbf{A} \triangleq \overline{\mathbf{U}}\overline{\boldsymbol{\Sigma}}\mathbf{C}$. The challenge is to determine the support of the optimal solution $\widehat{\mathbf{W}}$; if an oracle revealed the optimal supports $\mathcal{I}_j$, $j = 1, \ldots, k$ of its columns, then the exact value of the nonzero entries would be determined by the Cauchy-Schwarz inequality, and the contribution of the $j$th summand in (5) would be equal to $\sum_{i \in \mathcal{I}_j} A_{ij}^2$. Due to the non-negativity constrains in $\mathcal{W}_k$, the optimal support $\mathcal{I}_j$ of the $j$th column must contain indices corresponding to only nonnegative or nonpositive entries of $\mathbf{a}_j$, but not a combination of both. Algorithm 3 considers all $2^k$ possible sign combinations for the support sets implicitly by solving (5) on all $2^k$ matrices $\mathbf{A}' = \mathbf{A} \cdot \mathrm{diag}(\mathbf{s})$, $\mathbf{s} \in \{\pm 1\}^k$. Hence, we may assume without loss of generality that all support sets correspond to nonnegative entries of $\mathbf{A}$.

---

**Algorithm 3** `LocalOptW`

---

**input** : real $m \times k$ matrix $\mathbf{A}$
**output** $\widehat{\mathbf{W}} = \arg\max_{\mathbf{W} \in \mathcal{W}_k} \sum_{j=1}^{k} \left\langle \mathbf{w}_j, \, \mathbf{a}_j \right\rangle^2$
1: $\mathcal{C}_W \leftarrow \{\}$
2: **for each** $\mathbf{s} \in \{\pm 1\}^k$ **do**
3: $\quad \mathbf{A}' \leftarrow \mathbf{A} \cdot \mathrm{diag}(\mathbf{s})$
4: $\quad \mathcal{I}_j \leftarrow \{\}, \quad j = 1, \ldots, k$
5: $\quad$ **for** $i = 1 \ldots, m$ **do**
6: $\quad\quad j_\star \leftarrow \arg\max_j A'_{ij}$
7: $\quad\quad$ **if** $A'_{ij_\star} \geq 0$ **then**
8: $\quad\quad\quad \mathcal{I}_{j_\star} \leftarrow \mathcal{I}_{j_\star} \cup \{i\}$
9: $\quad\quad$ **end if**
10: $\quad$ **end for**
11: $\quad \mathbf{W} \leftarrow \mathbf{0}_{m \times k}$
12: $\quad$ **for** $j = 1, \ldots, k$ **do**
13: $\quad\quad [\mathbf{w}_j]_{\mathcal{I}_j} \leftarrow [\mathbf{a}'_j]_{\mathcal{I}_j} / \|[\mathbf{a}'_j]_{\mathcal{I}_j}\|$
14: $\quad$ **end for**
15: $\quad \mathcal{C}_W \leftarrow \mathcal{C}_W \cup \mathbf{W}$
16: **end for**
17: $\widehat{\mathbf{W}} \leftarrow \arg\max_{\mathbf{W} \in \mathcal{C}_W} \sum_{j=1}^{k} \left\langle \mathbf{w}_j, \, \mathbf{a}_j \right\rangle^2$

---

Moreover, if index $i \in [m]$ is assigned to $\mathcal{I}_j$, then the contribution of the entire $i$th row of $\mathbf{A}$ to the objective is equal to $A_{ij}^2$. Based on the above, Algorithm 3 constructs the collection of the support sets by assigning index $i$ to $\mathcal{I}_j$ if and only if $A_{ij}$ is nonnegative and the largest among the entries of the $i$th row of $\mathbf{A}$. The algorithm runs in time[2] $O(2^k \cdot k \cdot m)$ and guarantees that the output is the optimal solution to (5). A more formal analysis of the Alg. 3 is provided in Section A.5.

Thus far, we have seen that any given value of $\mathbf{C}$ can be associated with a feasible solution $\mathbf{W} \in \mathcal{W}_k$ via the maximization (5) and Alg. 3. If we could efficiently consider all possible values in the (continuous) domain of $\mathbf{C}$, we would be able to recover the pair that maximizes (4) and, in turn, the optimal solution of (3). However, that is not possible. Instead, we consider a fine discretization of the domain of $\mathbf{C}$ and settle for an approximate solution. In particular, let $\mathcal{N}_\epsilon(\mathbb{S}_2^{r-1})$ denote a finite $\epsilon$-net of the $r$-dimensional $\ell_2$-unit sphere; for any point in $\mathbb{S}_2^{r-1}$, the net contains a point within distance $\epsilon$ from the former. (see Appendix C for the construction of such a net). Further, let $[\mathcal{N}_\epsilon(\mathbb{S}_2^{r-1})]^{\otimes k}$ denote the $k$th Cartesian power of the previous net; the latter is a collection of $r \times k$ matrices $\mathbf{C}$. Alg. 1 operates on this collection: for each $\mathbf{C}$ it identifies a candidate solution $\mathbf{W} \in \mathcal{W}_k$ via the maximization (5) using Algorithm 3. By the properties of the $\epsilon$-nets, it can be shown that at least one of the computed candidate solutions must attain an objective value close to the optimal of (3).

The guarantees of Alg. 1 are formally established in Lemma 1. A detailed analysis of the algorithm is provided in the corresponding proof in Appendix A.2. This completes the description of our algorithmic developments.

## 4 Experimental Evaluation

**NNPCA** We compare our NNPCA algorithm against three existing approaches: NSPCA [16], EM [27] and NNSPAN [17] on real datasets. NSPCA computes multiple nonnegative, but not necessarily orthogonal components; a parameter $\alpha$ penalizes the overlap among their supports. We set a high penalty ($\alpha = 1e10$) to promote orthogonality. EM and NNSPAN compute only a single nonnegative component. Multiple components are computed consecutively, interleaving an appropriate deflation step. To ensure orthogonality, the deflation step effectively zeroes out the variables used in previously extracted components. Finally, note that both the EM and NSPCA algorithms are randomly initialized. All depicted values are the best results over multiple random restarts. For our algorithm, we use a sketch of rank $r = 4$ of the (centered) input data. Further we apply an early termination criterion; execution is terminated if no improvement is observed in a number of consecutive iterations (samples). This can only hurt the performance of our algorithm.

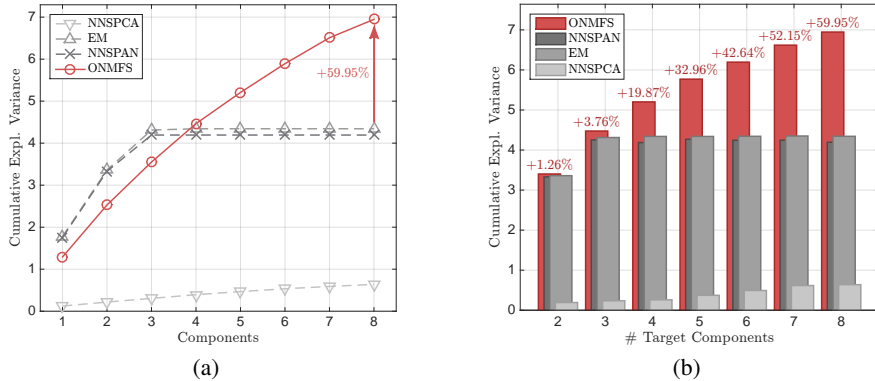

(a)                          (b)

Figure 2: Cumul. variance captured by $k$ nonnegative components; CBCL dataset [30]. In Fig. 2(a), we set $k = 8$ and plot the cumul. variance versus the number of components. EM and NNSPAN extract components greedily; first components achieve high value, but subsequent ones contribute less to the objective. Our algorithm jointly optimizes the $k = 8$ components, achieving a $59.95\%$ improvement over the second best method. Fig. 2(b) depicts the cumul. variance for various values of $k$. We note the percentage improvement of our algorithm over the second best method.

*CBCL Dataset.* The CBCL dataset [30] contains $2429$, $19 \times 19$ pixel, gray scale face images. It has been used in the evaluation of all three methods [16, 17, 27]. We extract $k$ orthogonal nonnegative components using all methods and compare the total explained variance, *i.e.*, the objective in (2). We note that input data has been centered and it is hence not nonnegative.

Fig. 2(a) depicts the cumulative explained variance versus the number of components for $k = 8$. EM and NNSPAN extract components greedily with a deflation step; the first component achieves high value, but subsequent ones contribute less to the total variance. On the contrary, our algorithm jointly optimizes the $k = 8$ components, achieving an approximately $60\%$ increase in the total variance compared to the second best method. We repeat the experiment for $k = 2, \ldots, 8$. Fig. 2(b) depicts the total variance captured by each method for each value of $k$. Our algorithm significantly outperforms the existing approaches.

*Additional Datasets.* We solve the NNPCA problem on various datasets obtained from [31]. We arbitrarily set the target number of components to $k = 5$ and configure our algorithm to use a rank-$4$ sketch of the input. Table 1 lists the total variance captured by the extracted components for each method. Our algorithm consistently outperforms the other approaches.

**ONMF** We compare our algorithm with several state-of-the-art ONMF algorithms *i)* the O-PNMF algorithm of [13] (for 1000 iterations), and *ii)* the more recent ONP-MF *iii)* EM-ONMF algorithms of [11, 32] (for 1000 iterations). We also compare to clustering methods (namely, vanilla and spherical $k$-means) since such algorithms also yield an approximate ONMF.

|  |  | NSPCA | EM | NNSPAN | ONMFS |
|---|---|---|---|---|---|
| AMZN COM. REV | ($1500 \times 10000$) | $5.44e + 01$ | $7.32e + 03$ | $7.32e + 03$ | $\mathbf{7.86e + 03}$ ($+7.37\%$) |
| ARCENCE TRAIN | ($100 \times 10000$) | $4.96e + 04$ | $3.01e + 07$ | $3.00e + 07$ | $\mathbf{3.80e + 07}$ ($+26.7\%$) |
| ISOLET-5 | ($1559 \times 617$) | $5.83e - 01$ | $3.54e + 01$ | $3.55e + 01$ | $\mathbf{4.55e + 01}$ ($+28.03\%$) |
| LEUKEMIA | ($72 \times 12582$) | $3.02e + 07$ | $7.94e + 09$ | $8.02e + 09$ | $\mathbf{1.04e + 10}$ ($+29.57\%$) |
| MFEAT PIX | ($2000 \times 240$) | $2.00e + 01$ | $3.20e + 02$ | $3.25e + 02$ | $\mathbf{5.24e + 02}$ ($+61.17\%$) |
| LOW RES. SPEC. | ($531 \times 100$) | $3.98e + 06$ | $2.29e + 08$ | $2.29e + 08$ | $\mathbf{2.41e + 08}$ ($+5.34\%$) |
| BOW:KOS | ($3430 \times 6906$) | $4.96e - 02$ | $2.96e + 01$ | $3.00e + 01$ | $\mathbf{4.59e + 01}$ ($+52.95\%$) |

Table 1: Total variance captured by $k = 5$ nonnegative components on various datasets [31]. For each dataset, we list (#samples$\times$#variables) and the variance captured by each method; higher values are better. Our algorithm (labeled ONMFS) operates on a rank-$4$ sketch in all cases, and consistently achieves the best results. We note the percentage improvement over the second best method.

*Synthetic data.* We generate a synthetic dataset as follows. We select five base vectors $\mathbf{c}_j$, $j = 1, \ldots, 5$ randomly and independently from the unit hypercube in 100 dimensions. Then, we generate data points $\mathbf{x}_i = a_i \cdot \mathbf{c}_j + p \cdot \mathbf{n}_i$, for some $j \in \{1, \ldots, 5\}$, where $a_i \sim U([0.1, 1])$, $\mathbf{n}_i \sim N(\mathbf{0}, \mathbf{I})$, and $p$ is a parameter controlling the noise variance. Any negative entries of $\mathbf{x}_i$ are set to zero.

We vary $p$ in $[10^{-2}, 1]$. For each $p$ value, we compute an approximate ONMF on 10 randomly generated datasets and measure the relative Frobenius approximation error. For the methods that involved random initialization, we run 10 averaging iterations per Monte-Carlo trial. Our algorithm is configured to operate on a rank-5 sketch. Figure 3 depicts the relative error achieved by each method (averaged over the random trials) versus the noise variance $p$. Our algorithm, labeled ONMFS achieves competitive or higher accuracy for most values in the range of $p$.

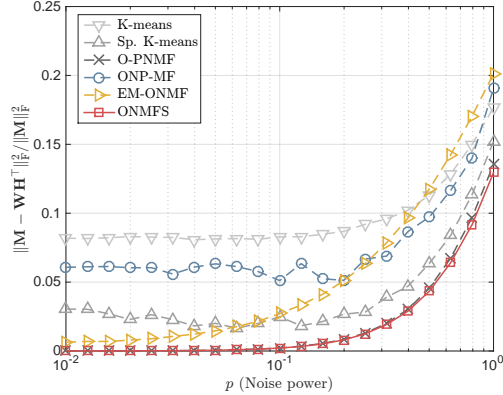

Figure 3: Relative Frob. approximation error on synthetic data. Data points (samples) are generated by randomly scaling and adding noise to one of five base points that have been randomly selected from the unit hypercube in 100 dimensions. We run ONMF methods with target dimension $k = 5$. Our algorithm is labeled as ONMFS.

*Real Datasets.* We apply the ONMF algorithms on various nonnegative datasets obtained from [31]. We arbitrarily set the target number of components to $k = 6$. Table 2 lists the relative Frobenius approximation error achieved by each algorithm. We note that on the text datasets (*e.g.*, Bag of Words [31]) we run the algorithms on the uncentered word-by-document matrix. Our algorithm performs competitively compared to other methods.

## 5   Conclusions

We presented a novel algorithm for approximately solving the ONMF problem on a nonnegative matrix. Our algorithm relied on a new method for solving the NNPCA problem. The latter jointly optimizes multiple orthogonal nonnegative components and provably achieves an objective value close to optimal. Our ONMF algorithm is the first one to be equipped with theoretical approximation guarantees; for a constant target dimension $k$, it yields an additive EPTAS for the relative approximation error. Empirical evaluation on synthetic and real datasets demonstrates that our algorithms outperform or match existing approaches in both problems.

**Acknowledgments**   DP is generously supported by NSF awards CCF-1217058 and CCF-1116404 and MURI AFOSR grant 556016. This research has been supported by NSF Grants CCF 1344179, 1344364, 1407278, 1422549 and ARO YIP W911NF-14-1-0258.

|  |  | K-MEANS | O-PNMF | ONP-MF | EM-ONMF | ONMFS |
|---|---|---|---|---|---|---|
| AMZN COM. REV | $(10000 \times 1500)$ | 0.0547 | 0.1153 | 0.1153 | 0.0467 | **0.0462**(5) |
| ARCENCE TRAIN | $(100 \times 10000)$ | 0.0837 | — | 0.1250 | 0.0856 | **0.0788**(4) |
| MFEAT PIX | $(2000 \times 240)$ | 0.2489 | 0.2974 | 0.3074 | **0.2447** | 0.2615 (4) |
| PEMS TRAIN | $(267 \times 138672)$ | 0.1441 | 0.1439 | 0.1380 | **0.1278** | 0.1283 (5) |
| BOW:KOS | $(3430 \times 6906)$ | 0.8193 | 0.7692 | 0.7671 | 0.7671 | **0.7609**(4) |
| BOW:ENRON | $(28102 \times 39861)$ | 0.9946 | — | 0.6728 | 0.7148 | **0.6540**(4) |
| BOW:NIPS | $(1500 \times 12419)$ | 0.8137 | 0.7277 | 0.7277 | 0.7375 | **0.7252**(5) |
| BOW:NYTIMES | $(102660 \times 3 \cdot 10^5)$ | — | — | **0.9199** | 0.9238 | **0.9199**(5) |

Table 2: ONMF approximation error on nonnegative datasets [31]. For each dataset, we list the size (#samples×#variables) and the relative Frobenius approximation error achieved by each method; lower values are better. We arbitrarily set the target dimension $k = 6$. Dashes (-) denote an invalid solution/non-convergence. For our method, we note in parentheses the approximation rank $r$ used.

## Footnotes

[1] Additive EPTAS (Efficient Polynomial Time approximation Scheme [28, 29]) refers to an algorithm that can approximate the solution of an optimization problem within an arbitrarily small additive error $\epsilon$ and has complexity that scales polynomially in the input size $n$, but possibly exponentially in $1/\epsilon$. EPTAS is more efficient than a PTAS because it enforces a polynomial dependency on $n$ for any $\epsilon$, *i.e.*, a running time $f(1/\epsilon) \cdot p(n)$, where $p(n)$ is polynomial. For example, a running time of $O(n^{1/\epsilon})$ is considered PTAS, but not EPTAS.

[2] When used as a subroutine in Alg. 1, Alg. 3 can be simplified into an $O(k \cdot m)$ procedure (lines 4-14).

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
