[Supplementary Material]

# Supplemental Material

## A Proofs

### A.1 On the connection of ONMF with NNPCA

**Lemma 2.** *Let $\mathcal{E}_\star \triangleq \min_{\mathbf{W} \geq \mathbf{0} \mathbf{H} \geq \mathbf{0}, \mathbf{W}^\top \mathbf{W} = \mathbf{I}_k} \|\mathbf{M} - \mathbf{W}\mathbf{H}^\top]\|_F^2$ be the optimal ONMF approximation error for a given $m \times n$ real nonnegative matrix $\mathbf{M}$ and target dimension $k$, as defined in* (1)*. Then,*

$$\mathcal{E}_\star = \|\mathbf{M}\|_F^2 - \max_{\substack{\mathbf{W} \geq \mathbf{0}_{m \times k} \\ \mathbf{W}^\top \mathbf{W} = \mathbf{I}_k}} \|\mathbf{M}^\top \mathbf{W}\|_F^2, \tag{6}$$

*If $\mathbf{W}_\star$ is a solution of the maximization in* (6)*, then the pair $\mathbf{W}_\star, \mathbf{H}_\star \triangleq \mathbf{M}^\top \mathbf{W}_\star$ is a feasible solution to the ONMF problem in* (1)*, achieving the minimum error $\mathcal{E}_\star$, i.e., $\|\mathbf{M} - \mathbf{W}_\star \mathbf{W}_\star^\top \mathbf{M}\|_F^2 = \mathcal{E}_\star$.*

*Proof.* Recall that by assumption, $\mathbf{W}$ is a $m \times k$ nonnegative matrix with orthonormal columns. The subsequent analysis holds even in the case where $\mathbf{W}$ is allowed to contain all-zero columns, as such columns do not contribute to the objective function and can be ignored effectively reducing the dimension $k$ of the factorization.

Given a real nonnegative $m \times k$ matrix $\mathbf{W}$, the real $n \times k$ matrix $\mathbf{H}$ that minimizes the Frobenius error $\|\mathbf{M} - \mathbf{W}\mathbf{H}^\top\|_F^2$ over all real $n \times k$ matrices (ignoring temporarily the fact what we seek a nonnegative $\mathbf{H}$), is given by $\mathbf{H}^\top = \mathbf{W}^\dagger \mathbf{M}$, where $\mathbf{W}^\dagger$ denotes the pseudo-inverse of $\mathbf{W}$. Here, however, the columns of $\mathbf{W}$ are orthonormal and hence $\mathbf{W}^\dagger = \mathbf{W}^\top$. Moreover, since $\mathbf{M}$ is nonnegative, $\mathbf{H}^\top = \mathbf{W}_\star^\dagger \mathbf{M} = \mathbf{W}_\star^\top \mathbf{M}$ automatically satisfies the additional nonnegativity constraint. Therefore, the ONMF problem (defined in (1)) reduces to a minimization in a single variable:

$$\mathcal{E}_\star = \min_{\substack{\mathbf{W} \geq \mathbf{0}_{m \times k} \\ \mathbf{W}^\top \mathbf{W} = \mathbf{I}_k}} \|\mathbf{M} - \mathbf{W}\mathbf{W}^\top \mathbf{M}\|_F^2. \tag{7}$$

Expanding the objective in (7),

$$\begin{aligned} \|\mathbf{M} - \mathbf{W}\mathbf{W}^\top \mathbf{M}\|_F^2 &= \|\mathbf{M}\|_F^2 - 2 \cdot \mathrm{Tr}\big(\mathbf{W}^\top \mathbf{M}\mathbf{M}^\top \mathbf{W}\big) + \mathrm{Tr}\big(\mathbf{M}^\top \mathbf{W}\mathbf{W}^\top \mathbf{M}\big) \\ &= \|\mathbf{M}\|_F^2 - \mathrm{Tr}\big(\mathbf{W}^\top \mathbf{M}\mathbf{M}^\top \mathbf{W}\big) \\ &= \|\mathbf{M}\|_F^2 - \|\mathbf{M}^\top \mathbf{W}\|_F^2, \end{aligned} \tag{8}$$

where the first step follows from the cyclic property of the trace and the fact that $\mathbf{W}^\top \mathbf{W} = \mathbf{I}_k$. This concludes the proof. $\qquad\square$

### A.2 Proof of Lemma 1

**Lemma 1.** *For any real $m \times n$ matrix $\overline{\mathbf{M}}$ with rank $r$, desired number of components $k$, and accuracy parameter $\epsilon \in (0, 1)$, Algorithm 1 outputs $\overline{\mathbf{W}} \in \mathcal{W}_k$ such that*

$$\|\overline{\mathbf{M}}^\top \overline{\mathbf{W}}\|_F^2 \geq (1 - \epsilon) \cdot \|\overline{\mathbf{M}}^\top \overline{\mathbf{W}}_\star\|_F^2,$$

*where $\overline{\mathbf{W}}_\star$ is the optimal solution defined in* (3)*, in time $T_{SVD} + O\big(\big(\frac{2}{\epsilon}\big)^{r \cdot k} \cdot k \cdot m\big)$.*

*Proof.* Let $\overline{\mathbf{M}} = \overline{\mathbf{U}}\,\overline{\mathbf{\Sigma}}\,\overline{\mathbf{V}}^\top$ denote the truncated eigenvalue decomposition of $\overline{\mathbf{M}}$; $\overline{\mathbf{\Sigma}}$ is a diagonal $r \times r$ matrix with $\Sigma_{ii}$ being equal to the $i$th largest singular value of $\overline{\mathbf{M}}$. For any $\mathbf{w} \in \mathbb{R}^m$,

$$\big\|\overline{\mathbf{M}}^\top \mathbf{w}\big\|_2^2 = \big\|\overline{\mathbf{V}}\,\overline{\mathbf{\Sigma}}\,\overline{\mathbf{U}}^\top \mathbf{w}\big\|_2^2 = \big\|\overline{\mathbf{\Sigma}}\,\overline{\mathbf{U}}^\top \mathbf{w}\big\|_2^2 \geq \big\langle \overline{\mathbf{\Sigma}}\,\overline{\mathbf{U}}^\top \mathbf{w}, \mathbf{c}\big\rangle^2, \quad \forall \, \mathbf{c} \in \mathbb{R}^r : \|\mathbf{c}\|_2 = 1, \tag{9}$$

where the first equality follows from the fact that the columns of $\overline{\mathbf{V}}$ are orthonormal and span the entire row space of $\overline{\mathbf{M}}$, and the inequality is due to Cauchy-Schwartz. In fact, equality is achieved for $\mathbf{c}$ colinear to $\overline{\mathbf{\Sigma}}\overline{\mathbf{U}}\mathbf{w}$, appropriately scaled to unit-length, and hence,

$$\big\|\overline{\mathbf{M}}^\top \mathbf{w}\big\|_2^2 = \max_{\mathbf{c} \in \mathbb{S}^{r-1}} \big\langle \overline{\mathbf{\Sigma}}\,\overline{\mathbf{U}}^\top \mathbf{w}, \mathbf{c}\big\rangle^2. \tag{10}$$

In turn,

$$\left\|\overline{\mathbf{M}}^\top \mathbf{W}\right\|_F^2 = \sum_{j=1}^k \left\|\overline{\mathbf{M}}^\top \mathbf{w}_j\right\|_2^2 = \sum_{j=1}^k \max_{\mathbf{c}_j \in \mathbb{S}^{r-1}} \left\langle \overline{\mathbf{\Sigma}}\overline{\mathbf{U}}^\top \mathbf{w}_j,\, \mathbf{c}_j \right\rangle^2. \tag{11}$$

Recall that $\overline{\mathbf{W}}_\star$ by definition maximizes the left hand side of (11) over all $\mathbf{W} \in \mathcal{W}_k$. Let $\widetilde{\mathbf{c}}_{\star 1}, \ldots, \widetilde{\mathbf{c}}_{\star k} \in \mathbb{S}^{r-1}$ be the set of $k$ vectors achieving equality in (11) for $\mathbf{W} = \overline{\mathbf{W}}_\star$, and let $\widetilde{\mathbf{C}}_\star \in \mathbb{R}^{r \times k}$ be the matrix formed by stacking the $k$ vectors. Algorithm 1 iterates over a set $\mathcal{N}_{\epsilon/2}^{\otimes k}\left(\mathbb{S}^{r-1}\right)$ of points ($r \times k$ matrices) $\mathbf{C}$. Recall that $\mathcal{N}_{\epsilon/2}^{\otimes k}\left(\mathbb{S}^{r-1}\right)$ is the $k$th cartesian power of an $\epsilon/2$-net of the $r$-dimensional $\ell_2$-unit sphere. By construction, the set contains a matrix $\mathbf{C}_\sharp$ such that

$$\|\mathbf{C}_\sharp - \widetilde{\mathbf{C}}_{\star j}\|_{\infty,2} \le \epsilon/2. \tag{12}$$

Then, for all $j \in \{1, \ldots, k\}$,

$$\begin{aligned}
\left\|\overline{\mathbf{M}}^\top \overline{\mathbf{W}}_{\star j}\right\|_2 &= \left|\left\langle \overline{\mathbf{\Sigma}}\overline{\mathbf{U}}^\top \overline{\mathbf{W}}_{\star j},\, \widetilde{\mathbf{c}}_{\star j} \right\rangle\right| \\
&= \left|\left\langle \overline{\mathbf{\Sigma}}\overline{\mathbf{U}}^\top \overline{\mathbf{W}}_{\star j},\, \mathbf{c}_{\sharp j} \right\rangle + \left\langle \overline{\mathbf{\Sigma}}\overline{\mathbf{U}}^\top \overline{\mathbf{W}}_{\star j},\, (\widetilde{\mathbf{c}}_{\star j} - \mathbf{c}_{\sharp j}) \right\rangle\right| \\
&\le \left|\left\langle \overline{\mathbf{\Sigma}}\overline{\mathbf{U}}^\top \overline{\mathbf{W}}_{\star j},\, \mathbf{c}_{\sharp j} \right\rangle\right| + \left|\left\langle \overline{\mathbf{\Sigma}}\overline{\mathbf{U}}^\top \overline{\mathbf{W}}_{\star j},\, (\widetilde{\mathbf{c}}_{\star j} - \mathbf{c}_{\sharp j}) \right\rangle\right| \\
&\le \left|\left\langle \overline{\mathbf{\Sigma}}\overline{\mathbf{U}}^\top \overline{\mathbf{W}}_{\star j},\, \mathbf{c}_{\sharp j} \right\rangle\right| + \left\|\overline{\mathbf{\Sigma}}\overline{\mathbf{U}}^\top \overline{\mathbf{W}}_{\star j}\right\|_2 \cdot \|\widetilde{\mathbf{c}}_{\star j} - \mathbf{c}_{\sharp j}\|_2 \\
&\le \left|\left\langle \overline{\mathbf{\Sigma}}\overline{\mathbf{U}}^\top \overline{\mathbf{W}}_{\star j},\, \mathbf{c}_{\sharp j} \right\rangle\right| + (\epsilon/2) \cdot \left\|\overline{\mathbf{M}}^\top \overline{\mathbf{W}}_{\star j}\right\|_2.
\end{aligned} \tag{13}$$

The first step follows by the definition of $\widetilde{\mathbf{C}}_\star$, the second by the linearity of the inner product, the third by the triangle inequality, the fourth by Cauchy-Schwarz inequality and the last by the fact that $\|\widetilde{\mathbf{c}}_{\star j} - \mathbf{c}_{\sharp j}\| \le \epsilon/2, \forall i \in \{1, \ldots, k\}$ (by (12)). Rearranging the terms in (13),

$$\left|\left\langle \overline{\mathbf{\Sigma}}\overline{\mathbf{U}}^\top \overline{\mathbf{W}}_{\star j},\, \mathbf{c}_{\sharp j} \right\rangle\right| \ge \left(1 - \tfrac{\epsilon}{2}\right) \cdot \left\|\overline{\mathbf{M}}^\top \overline{\mathbf{W}}_{\star j}\right\|_2 \ge 0,$$

which in turn implies (by taking the square on both sides) that

$$\left\langle \overline{\mathbf{\Sigma}}\overline{\mathbf{U}}^\top \overline{\mathbf{W}}_{\star j},\, \mathbf{c}_{\sharp j} \right\rangle^2 \ge \left\|\overline{\mathbf{M}}^\top \overline{\mathbf{W}}_{\star j}\right\|_2^2 \ge (1 - \epsilon) \cdot \left\|\overline{\mathbf{M}}^\top \overline{\mathbf{W}}_{\star j}\right\|_2^2 \tag{14}$$

Summing the terms in (14) over all $j \in \{1, \ldots, k\}$,

$$\sum_{j=1}^k \left\langle \overline{\mathbf{\Sigma}}\overline{\mathbf{U}}^\top \overline{\mathbf{W}}_{\star j},\, \mathbf{c}_{\sharp j} \right\rangle^2 \ge (1 - \epsilon) \cdot \left\|\overline{\mathbf{M}}^\top \overline{\mathbf{W}}_\star\right\|_F^2. \tag{15}$$

Let $\mathbf{W}_\sharp \in \mathcal{W}_k$ be the candidate solution produced by the algorithm at $\mathbf{C}_\sharp$, i.e.,

$$\mathbf{W}_\sharp \triangleq \arg\max_{\mathbf{W} \in \mathcal{W}_k} \sum_{j=1}^k \left\langle \mathbf{w}_j,\, \overline{\mathbf{U}}\overline{\mathbf{\Sigma}}\mathbf{c}_{\sharp j} \right\rangle^2 \tag{16}$$

Then,

$$\begin{aligned}
\left\|\overline{\mathbf{M}}^\top \mathbf{W}_\sharp\right\| &\overset{(\alpha)}{=} \sum_{j=1}^k \max_{\mathbf{c}_j \in \mathbb{S}^{r-1}} \left\langle \overline{\mathbf{\Sigma}}\overline{\mathbf{U}}^\top \overline{\mathbf{W}}_{\sharp j},\, \mathbf{c}_j \right\rangle^2 \\
&\overset{(\beta)}{\ge} \sum_{j=1}^k \left\langle \overline{\mathbf{\Sigma}}\overline{\mathbf{U}}^\top \overline{\mathbf{W}}_{\sharp j},\, \mathbf{c}_{\sharp j} \right\rangle^2 \\
&\overset{(\gamma)}{\ge} \sum_{j=1}^k \left\langle \overline{\mathbf{W}}_{\star j},\, \overline{\mathbf{U}}\overline{\mathbf{\Sigma}}\mathbf{c}_{\sharp j} \right\rangle^2 \\
&\overset{(\delta)}{\ge} (1 - \epsilon) \cdot \left\|\overline{\mathbf{M}}\overline{\mathbf{W}}_\star\right\|_F^2,
\end{aligned} \tag{17}$$

where ($\alpha$) follows from the observation in (11), ($\beta$) from the suboptimality of $\mathbf{C}_\sharp$, ($\gamma$) from the fact that $\mathbf{W}_\sharp$ maximizes the sum by its definition in (16), while ($\delta$) follows from (15). According to (17), at least one of the candidate solutions produced by Algorithm 1, namely $\mathbf{W}_\sharp$, achieves an objective value within a multiplicative factor $(1-\epsilon)$ from the optimal, implying the guarantees of the lemma.

Finally, the running time of Algorithm 1 follows immediately from the cost per iteration and the cardinality of the $\epsilon/2$-net on the unit-sphere. Note that matrix multiplications can exploit the available singular value decomposition which is performed once. $\qquad\square$

## A.3   Proof of Theorem 1

We first prove some auxiliary lemmata. The proof of the Theorem is given in the end of this section.

**Lemma 3.** *For any real $m \times n$ matrices $\mathbf{M}$ and $\overline{\mathbf{M}}$, let*

$$\mathbf{W}_\star \triangleq \underset{\mathbf{W} \in \mathcal{W}_k}{\arg\max} \big\|\mathbf{M}^\top \mathbf{W}\big\|_{\mathrm{F}}^2 \quad and \quad \overline{\mathbf{W}}_\star \triangleq \underset{\mathbf{W} \in \mathcal{W}_k}{\arg\max} \big\|\overline{\mathbf{M}}^\top \mathbf{W}\big\|_{\mathrm{F}}^2, \tag{18}$$

*respectively. Then, for any $\overline{\mathbf{W}} \in \mathcal{W}_k$ such that $\big\|\overline{\mathbf{M}}^\top \overline{\mathbf{W}}\big\|_{\mathrm{F}}^2 \geq \gamma \cdot \big\|\overline{\mathbf{M}}^\top \overline{\mathbf{W}}_\star\big\|_{\mathrm{F}}^2$ for some $0 < \gamma < 1$,*

$$\big\|\mathbf{M}^\top \overline{\mathbf{W}}\big\|_{\mathrm{F}}^2 \geq \gamma \cdot \big\|\mathbf{M}^\top \mathbf{W}_\star\big\|_{\mathrm{F}}^2 - 2 \cdot k \cdot \|\mathbf{M} - \overline{\mathbf{M}}\|_2^2.$$

*Proof.* By the optimality of $\overline{\mathbf{W}}_\star$ for $\overline{\mathbf{M}}$,

$$\big\|\overline{\mathbf{M}}^\top \overline{\mathbf{W}}_\star\big\|_{\mathrm{F}}^2 \geq \big\|\overline{\mathbf{M}}^\top \mathbf{W}_\star\big\|_{\mathrm{F}}^2.$$

In turn, for any $\overline{\mathbf{W}} \in \mathcal{W}_k$ satisfying the assumptions of the lemma,

$$\big\|\overline{\mathbf{M}}^\top \overline{\mathbf{W}}\big\|_{\mathrm{F}}^2 \geq \gamma \cdot \big\|\overline{\mathbf{M}}^\top \mathbf{W}_\star\big\|_{\mathrm{F}}^2. \tag{19}$$

Let $\mathbf{A} \triangleq \mathbf{M}\mathbf{M}^\top$, $\widetilde{\mathbf{A}} \triangleq \overline{\mathbf{M}}\overline{\mathbf{M}}^\top$, and $\mathbf{E} \triangleq \mathbf{A} - \widetilde{\mathbf{A}}$. By the linearity of the trace,

$$\begin{aligned}
\big\|\overline{\mathbf{M}}^\top \overline{\mathbf{W}}\big\|_{\mathrm{F}}^2 &= \mathrm{TR}\big(\overline{\mathbf{W}}^\top \mathbf{A} \overline{\mathbf{W}}\big) - \mathrm{TR}\big(\overline{\mathbf{W}}^\top \mathbf{E} \overline{\mathbf{W}}\big) \\
&\leq \mathrm{TR}\big(\overline{\mathbf{W}}^\top \mathbf{A} \overline{\mathbf{W}}\big) + \big|\mathrm{TR}\big(\overline{\mathbf{W}}^\top \mathbf{E} \overline{\mathbf{W}}\big)\big|.
\end{aligned} \tag{20}$$

By Lemma 10,

$$\big|\mathrm{TR}\big(\overline{\mathbf{W}}^\top \mathbf{E} \overline{\mathbf{W}}\big)\big| \leq \|\overline{\mathbf{W}}\|_{\mathrm{F}}^2 \cdot \|\mathbf{E}\|_2 \leq k \cdot \|\mathbf{E}\|_2 \triangleq R, \tag{21}$$

where the last inequality follows from the fact that $\|\mathbf{W}\|_{\mathrm{F}}^2 \leq k$ for any $\mathbf{W} \in \mathcal{W}_k$. Continuing from (20),

$$\big\|\overline{\mathbf{M}}^\top \overline{\mathbf{W}}\big\|_{\mathrm{F}}^2 \leq \mathrm{TR}\big(\overline{\mathbf{W}}^\top \mathbf{A} \overline{\mathbf{W}}\big) + R. \tag{22}$$

Similarly,

$$\begin{aligned}
\big\|\overline{\mathbf{M}}^\top \mathbf{W}_\star\big\|_{\mathrm{F}}^2 &= \mathrm{TR}\big(\mathbf{W}_\star^\top \mathbf{A} \mathbf{W}_\star\big) - \mathrm{TR}\big(\mathbf{W}_\star^\top \mathbf{E} \mathbf{W}_\star\big) \\
&\geq \mathrm{TR}\big(\mathbf{W}_\star^\top \mathbf{A} \mathbf{W}_\star\big) - \big|\mathrm{TR}\big(\mathbf{W}_\star^\top \mathbf{E} \mathbf{W}_\star\big)\big| \\
&\geq \mathrm{TR}\big(\mathbf{W}_\star^\top \mathbf{A} \mathbf{W}_\star\big) - R.
\end{aligned} \tag{23}$$

Combining the above, we have

$$\begin{aligned}
\mathrm{TR}\big(\overline{\mathbf{W}}^\top \mathbf{A} \overline{\mathbf{W}}\big) &\geq \big\|\overline{\mathbf{M}}^\top \overline{\mathbf{W}}\big\|_{\mathrm{F}}^2 - R \\
&\geq \gamma \cdot \big\|\overline{\mathbf{M}}^\top \mathbf{W}_\star\big\|_{\mathrm{F}}^2 - R \\
&\geq \gamma \cdot \big(\mathrm{TR}\big(\mathbf{W}_\star^\top \mathbf{A} \mathbf{W}_\star\big) - R\big) - R \\
&= \gamma \cdot \mathrm{TR}\big(\mathbf{W}_\star^\top \mathbf{A} \mathbf{W}_\star\big) - (1 + \gamma) \cdot R \\
&\geq \gamma \cdot \mathrm{TR}\big(\mathbf{W}_\star^\top \mathbf{A} \mathbf{W}_\star\big) - 2 \cdot R,
\end{aligned}$$

where the first inequality follows from (22) the second from (19), the third from (23), and the last from the fact that $R \geq 0$ and $0 < \gamma \leq 1$. This concludes the proof. $\qquad\square$

**Remark 1.** *If in Lemma 3 $\overline{\mathbf{M}}$ is such that $\mathbf{MM}^\top - \overline{\mathbf{M}}\,\overline{\mathbf{M}}^\top$ is PSD, then*

$$\left\|\mathbf{M}^\top\overline{\mathbf{W}}\right\|_{\mathrm{F}}^2 \geq \gamma \cdot \left\|\mathbf{M}^\top\mathbf{W}_\star\right\|_{\mathrm{F}}^2 - \sum_{i=1}^{k}\lambda_i\big(\mathbf{MM}^\top - \overline{\mathbf{M}}\,\overline{\mathbf{M}}^\top\big).$$

*Proof.* This follows from the fact that if $\mathbf{E} \triangleq \mathbf{A} - \widetilde{\mathbf{A}}$ is PSD, then

$$\mathrm{TR}\left(\widetilde{\mathbf{X}}^\top\mathbf{E}\widetilde{\mathbf{X}}\right) = \sum_{j}^{m}\mathbf{x}_j^\top\mathbf{E}\mathbf{x}_j \geq 0,$$

and the bound in (20) can be improved to

$$\left\|\overline{\mathbf{M}}^\top\overline{\mathbf{W}}\right\|_{\mathrm{F}}^2 = \mathrm{TR}\left(\widetilde{\mathbf{X}}^\top\mathbf{A}\widetilde{\mathbf{X}}\right) - \mathrm{TR}\left(\widetilde{\mathbf{X}}^\top\mathbf{E}\widetilde{\mathbf{X}}\right)$$
$$\leq \mathrm{TR}\left(\widetilde{\mathbf{X}}^\top\mathbf{A}\widetilde{\mathbf{X}}\right).$$

Further, by Lemma 10 (Corollary 3) the bound in (21) becomes

$$\mathrm{TR}\left(\overline{\mathbf{W}}^\top\mathbf{E}\overline{\mathbf{W}}\right) \leq \|\overline{\mathbf{W}}\|_{\mathrm{F}}^2 \cdot \|\mathbf{E}\|_2 \leq \sum_{i=1}^{k}\lambda_i(\mathbf{E}).$$

The rest of the proof follows. $\qquad\square$

**Theorem 1.** *For any real $m \times n$ (not necessarily nonnegative) matrix $\mathbf{M}$ and desired number of components $k$, let $\mathbf{W}_\star \triangleq \arg\max_{\mathbf{W}\in\mathcal{W}_k}\left\|\mathbf{M}^\top\mathbf{W}\right\|_{\mathrm{F}}^2$. Let $\overline{\mathbf{M}}$ be the best rank-$r$ approximation of $\mathbf{M}$. Algorithm 1 with input $\overline{\mathbf{M}}$ and accuracy parameters $\epsilon$ and $r$, outputs $\overline{\mathbf{W}} \in \mathcal{W}_k$ such that*

$$\left\|\mathbf{M}^\top\overline{\mathbf{W}}\right\|_{\mathrm{F}}^2 \geq (1-\epsilon)\cdot\left\|\mathbf{M}^\top\mathbf{W}_\star\right\|_{\mathrm{F}}^2 - k\cdot\|\mathbf{M}-\overline{\mathbf{M}}\|_2^2$$

*in time $T_{\mathsf{SVD}} + O\left(\left(\frac{1}{\epsilon}\right)^{r\cdot k}\cdot k \cdot m\right).$*

*Proof.* Let $\overline{\mathbf{W}}$ be the output of Algorithm 1 with input the best rank-$r$ approximation of of $\mathbf{M}$, $\overline{\mathbf{M}}$. By the guarantees of Algorithm 1, (Lemma 1), the output $\overline{\mathbf{W}} \in \mathcal{W}_k$ of Algorithm 1 is such that

$$\left\|\overline{\mathbf{M}}^\top\overline{\mathbf{W}}\right\|_{\mathrm{F}}^2 \geq (1-\epsilon)\cdot\left\|\overline{\mathbf{M}}^\top\overline{\mathbf{W}}_\star\right\|_{\mathrm{F}}^2,$$

where $\overline{\mathbf{W}}_\star \triangleq \arg\max_{\mathbf{W}\in\mathcal{W}_k}\|\overline{\mathbf{M}}^\top\mathbf{W}\|_{\mathrm{F}}^2$. In turn, by Lemma 3 (and in particular taking into account the remark 1 whose conditions are satisfied since $\mathbf{MM} - \overline{\mathbf{M}}\,\overline{\mathbf{M}}^\top$ is PSD), we have

$$\left\|\mathbf{M}^\top\overline{\mathbf{W}}\right\|_{\mathrm{F}}^2 \geq (1-\epsilon)\cdot\left\|\mathbf{M}^\top\mathbf{W}_\star\right\|_{\mathrm{F}}^2 - \sum_{i=1}^{k}\lambda_i\big(\mathbf{MM}^\top - \overline{\mathbf{M}}\,\overline{\mathbf{M}}^\top\big)$$

$$= (1-\epsilon)\cdot\left\|\mathbf{M}^\top\mathbf{W}_\star\right\|_{\mathrm{F}}^2 - \sum_{i=r+1}^{r+k}\sigma_i^2(\mathbf{M}). \qquad (24)$$

The desired result readily follows. $\qquad\square$

## A.4 Proof of Theorem 2

**Lemma 4.** *For any $m \times n$ real nonnegative matrix $\mathbf{M}$, target dimension $k$, and accuracy parameters $r \in [n]$ and $\epsilon > 0$, Algorithm 2 outputs an ONMF pair $\overline{\mathbf{W}}, \overline{\mathbf{H}}$, such that*

$$\left\|\mathbf{M} - \overline{\mathbf{W}}\,\overline{\mathbf{H}}^\top\right\|_{\mathrm{F}}^2 \leq \mathcal{E}_\star + \epsilon\cdot\sum_{i=1}^{k}\sigma_i^2(\mathbf{M}) + \sum_{i=r+1}^{r+k}\sigma_i^2(\mathbf{M}),$$

*in time $T_{\mathsf{SVD}} + O\left(\left(\frac{2}{\epsilon}\right)^{r\cdot k}\cdot k \cdot m\right).$*

*Proof.* Recall that given a real nonnegative $m \times k$ matrix $\mathbf{W} \in \mathcal{W}_k$, $\mathbf{H}^\top = \mathbf{W}_\star^\top \mathbf{M}$ minimizes the Frobenius error $\|\mathbf{M} - \mathbf{W}\mathbf{H}^\top\|_F^2$ over the set of real nonnegative $n \times k$ matrices (Proof of Lemma 2). In turn, for any $\mathbf{W} \in \mathcal{W}_k$, and $\mathbf{H}$ selected as above,

$$\|\mathbf{M} - \mathbf{W}\mathbf{H}^\top\|_F^2 = \|\mathbf{M}\|_F^2 - \|\mathbf{M}^\top \mathbf{W}\|_F^2. \tag{25}$$

Let $\overline{\mathbf{W}}$ be the output of Algorithm 1, for input matrix $\mathbf{M}_r$ the best rank-$r$ approximation of of $\mathbf{M}$. That is, in the sequel of this proof, $\overline{\mathbf{M}} = \mathbf{M}_r$. By the guarantees of Algorithm 1, (Lemma 1), the output $\overline{\mathbf{W}} \in \mathcal{W}_k$ of Algorithm 1 is such that

$$\|\mathbf{M}_r^\top \overline{\mathbf{W}}\|_F^2 \geq (1 - \epsilon) \cdot \|\mathbf{M}_r^\top \overline{\mathbf{W}}_\star\|_F^2,$$

where $\overline{\mathbf{W}}_\star \triangleq \arg\max_{\mathbf{W} \in \mathcal{W}_k} \|\mathbf{M}_r^\top \mathbf{W}\|_F^2$. In turn, by Lemma 3 (and in particular taking into account the remark 1 whose conditions are satisfied since $\mathbf{M}\mathbf{M} - \mathbf{M}_r\mathbf{M}_r^\top$ is PSD), we have

$$\left\|\mathbf{M}^\top \overline{\mathbf{W}}\right\|_F^2 \geq (1 - \epsilon) \cdot \left\|\mathbf{M}^\top \mathbf{W}_\star\right\|_F^2 - \sum_{i=1}^{k} \lambda_i\big(\mathbf{M}\mathbf{M}^\top - \mathbf{M}_r\mathbf{M}_r^\top\big)$$

$$= (1 - \epsilon) \cdot \left\|\mathbf{M}^\top \mathbf{W}_\star\right\|_F^2 - \sum_{i=r+1}^{r+k} \sigma_i^2(\mathbf{M}). \tag{26}$$

Given the output $\overline{\mathbf{W}}$ of Algorithm 1, Algorithm 2 outputs the pair $\overline{\mathbf{W}}, \overline{\mathbf{H}}^\top \triangleq \overline{\mathbf{W}}^\top \mathbf{M}$. By (25), for this choice of $\overline{\mathbf{H}}$, taking into account (26),

$$\|\mathbf{M} - \overline{\mathbf{W}}\overline{\mathbf{H}}^\top\|_F^2 = \|\mathbf{M}\|_F^2 - \|\mathbf{M}^\top \overline{\mathbf{W}}\|_F^2$$

$$\leq \|\mathbf{M}\|_F^2 - (1 - \epsilon) \cdot \left\|\mathbf{M}^\top \mathbf{W}_\star\right\|_F^2 + \sum_{i=r+1}^{r+k} \sigma_i^2(\mathbf{M})$$

$$= \|\mathbf{M}\|_F^2 - \left\|\mathbf{M}^\top \mathbf{W}_\star\right\|_F^2 + \epsilon \cdot \left\|\mathbf{M}^\top \mathbf{W}_\star\right\|_F^2 + \sum_{i=r+1}^{r+k} \sigma_i^2(\mathbf{M})$$

$$= \|\mathbf{M} - \mathbf{W}_\star \mathbf{H}_\star^\top\|_F^2 + \epsilon \cdot \left\|\mathbf{M}^\top \mathbf{W}_\star\right\|_F^2 + \sum_{i=r+1}^{r+k} \sigma_i^2(\mathbf{M})$$

$$= \|\mathbf{M} - \mathbf{W}_\star \mathbf{H}_\star^\top\|_F^2 + \epsilon \cdot \sum_{i=1}^{k} \sigma_i^2(\mathbf{M}) + \sum_{i=r+1}^{r+k} \sigma_i^2(\mathbf{M}),$$

where the last inequality follows by Lemma 10. This competes the proof. $\square$

**Theorem 2.** *For any $m \times n$ real nonnegative matrix $\mathbf{M}$, target dimension $k$, and desired accuracy $0 < \epsilon < 1$, Algorithm 2 with parameters $\epsilon$ and $r = \lceil k/\epsilon \rceil$ outputs an ONMF pair $\mathbf{W}, \mathbf{H}$, such that*

$$\|\mathbf{M} - \mathbf{W}\mathbf{H}^\top\|_F^2 \ \leq \ \mathcal{E}_\star + \varepsilon \cdot \|\mathbf{M}\|_F^2,$$

*in time $T_{SVD} + \left(\frac{1}{\epsilon}\right)^{(k^2/\epsilon)} \cdot (k \cdot m)$.*

*Proof.* By Lemma 4, Algorithm 2 with parameters $r$ and $\epsilon$, outputs an ONMF pair $\overline{\mathbf{W}}, \overline{\mathbf{H}}$, such that

$$\|\mathbf{M} - \overline{\mathbf{W}}\overline{\mathbf{H}}^\top\|_F^2 \leq \mathcal{E}_\star + \epsilon \cdot \sum_{i=1}^{k} \sigma_i^2(\mathbf{M}) + \sum_{i=r+1}^{r+k} \sigma_i^2(\mathbf{M}). \tag{27}$$

Noting that for $k < r$ the $k$ (squared) singular values $\sigma_i^2(\mathbf{M})$, $i = r+1, \ldots, r+k$ are the smallest among the $r$ (squared) singular values $\sigma_i^2(\mathbf{M})$, $i = k+1, \ldots, r+k$, the last term in the right-hand side can be upper bounded as follows:

$$\sum_{i=r+1}^{r+k} \sigma_i^2(\mathbf{M}) \leq \frac{k}{r} \cdot \sum_{i=k+1}^{r+k} \sigma_i^2(\mathbf{M}). \tag{28}$$

For $r = \lceil k/\epsilon \rceil$, and combining the (28) and (27), we have

$$\|\mathbf{M} - \overline{\mathbf{W}}\overline{\mathbf{H}}^\top\|_F^2 \leq \mathcal{E}_\star + \epsilon \cdot \sum_{i=1}^{k} \sigma_i^2(\mathbf{M}) + \epsilon \cdot \sum_{i=k+1}^{r+k} \sigma_i^2(\mathbf{M})$$

$$= \mathcal{E}_\star + \epsilon \cdot \sum_{i=1}^{r+k} \sigma_i^2(\mathbf{M})$$

$$\leq \mathcal{E}_\star + \epsilon \cdot \|\mathbf{M}\|_F^2,$$

which is the desired guarantee. The time complexity readily follows from the steps of Algorithm 2 and that of Algorithm 1, which concludes the proof. $\qquad\square$

### A.5 Correctness of Algorithm 3

**Lemma 5.** *For any $m \times k$ matrix $\mathbf{A}$, Algorithm 3 outputs the $m \times k$ nonnegative matrix*

$$\widehat{\mathbf{W}} = \arg\max_{\mathbf{W} \in \mathcal{W}_k} \sum_{j=1}^{k} \langle \mathbf{w}_j, \, \mathbf{a}_j \rangle^2,$$

*in time $O(2^k \cdot k \cdot m)$.*

*Proof.* Let $\mathcal{I}_j \subseteq [m]$, $j = 1 \ldots, k$ denote the supports of the $k$ columns of optimal solution $\widehat{\mathbf{W}}$. The orthogonality requirements (in conjunction with nonnegativity) imply that the supports $\mathcal{I}_j$, $j = 1, \ldots, k$ are disjoint. Further, it is straightforward to verify that due to the nonnegativity constrains in $\mathcal{W}_k$, the support of the $j$th column, $\mathcal{I}_j$, must contain only indices corresponding to nonnegative or nonpositive entries of $\mathbf{a}_j$, but not a combination of both. Algorithm 3 considers all $2^k$ sign combinations for the support sets, *e.g.*, $\mathcal{I}_1$ containing positive entries, $\mathcal{I}_2$ negative, etc., by equivalently solving the maximization on all $2^k$ matrices $\hat{\mathbf{A}} = \mathbf{A} \cdot \mathrm{diag}(\mathbf{s})$, $b \in \{\pm 1\}^k$ and returning the solution that performs best on the original input $\mathbf{A}$. Therefore, without loss of generality, in the sequel we assume that all support sets correspond to nonnegative entries of $\mathbf{A}$.

If an oracle reveals the supports $\mathcal{I}_j$, $j = 1, \ldots, k$, the exact value of $\widehat{\mathbf{X}}$ can be readily determined, according to the Cauchy-Schwarz inequality: the $j$th column, $\hat{\mathbf{x}}_j$, is supported only on $\mathcal{I}_j$, and its nonzero sub-vector is set to $(\hat{\mathbf{x}}_j)_{\mathcal{I}_j} = [\mathbf{a}_j]_{\mathcal{I}_j} / \|[\mathbf{a}_j]_{\mathcal{I}_j}\|$, which maximizes the inner product with the corresponding sub-vector of $\mathbf{a}_j$. In turn, the objective function attains value equal to

$$\sum_{j=1}^{k} (\hat{\mathbf{x}}_j^\top \mathbf{a}_j)^2 = \sum_{j=1}^{k} \sum_{i \in \mathcal{I}_j} A_{ij}^2, \tag{29}$$

where the first equality stems from the fact that $[\mathbf{a}_j]_{\mathcal{I}_j} \geq \mathbf{0}$. It suffices to show that Alg. 3 correctly determines the collection of support sets $\mathcal{I}_j$, $j = 1, \ldots, k$.

Alg. 3 constructs the collection of support sets $\mathcal{I}_j$, $j = 1, \ldots, k$, according to the following rule:

$$i \in \mathcal{I}_j \Leftrightarrow A_{ij} > \max\{0, \, A_{iw}\}, \, \forall w \in [k] \backslash \{j\}, \tag{30}$$

*i.e.*, index $i \in [m]$ is assigned to the support of the $j$th column if and only if $A_{ij}$ is positive and the largest entry in the $i$th row of $\mathbf{A}$. Note that any procedure to construct supports that satisfy the requirements described in the beginning of this proof would assign each index $i \in [m]$ to at most one of the sets $\mathcal{I}_j$, $j \in \{1, \ldots, k\}$, while it would need to ensure that $i \in \mathcal{I}_j$ if and only if $A_{ij} > 0$. The rule in (30) additionally requires that index $i \in [m]$ is assigned to $\mathcal{I}_j$ if and only if $A_{ij}$ is the *largest* (positive) entry in the $i$th row of $\mathbf{A}$.

Assume, for the sake of contradiction, that there exists a set of optimal supports $\mathcal{I}_j$, $j = 1, \ldots, k$ which does not adhere to the rule in (30), *i.e.*, there exist $u \in [k]$ and $q \in [m]$, such $q \in \mathcal{I}_u$, while $0 < A_{qu} < A_{qv}$ for some $v \in [k]$, $v \neq u$. Consider a collection of supports $\widetilde{\mathcal{I}}_j$, $j = 1, \ldots, k$, with

$$\widetilde{\mathcal{I}}_j = \mathcal{I}_j, \, \forall j \in [k] \backslash \{u, v\}, \quad \widetilde{\mathcal{I}}_u = \mathcal{I}_u \backslash \{q\} \quad \text{and} \quad \widetilde{\mathcal{I}}_v = \mathcal{I}_v \cup \{q\}. \tag{31}$$

Note that the collection of supports in (31) satisfies the desired constraints. Further, the objective value achieved for the new supports (according to (29)) is equal to

$$\sum_{j=1}^{k}\sum_{i\in\widetilde{\mathcal{I}}_j} A_{ij}^2 = \sum_{j=1}^{k}\sum_{i\in\mathcal{I}_j} A_{ij}^2 - A_{qu}^2 + A_{qv}^2 > \sum_{j=1}^{k}\sum_{i\in\mathcal{I}_j} A_{ij}^2,$$

contradicting the optimality of the collection $\mathcal{I}_j$, $j = 1,\dots,k$. We conclude that the collection of optimal support sets must satisfy (30).

The construction of the support sets according to (30) requires determining the largest entry of each of the $m$ rows of $\mathbf{A}$, which can be done in $O(km)$. Once the supports are determined, each of the $k$ columns of $\widehat{\mathbf{X}}$ is constructed in $O(m)$. Taking into account that the above procedure is repeated $2^k$ times for each of the sign patters, the desired result follows. □

## B  Auxiliary Technical Results

**Lemma 6.** *For any real $m \times n$ matrix $\mathbf{M}$, and any $r, k \le \min\{m, n\}$,*

$$\sum_{i=r+1}^{r+k} \sigma_i(\mathbf{M}) \le \frac{k}{\sqrt{r+k}} \cdot \|\mathbf{M}\|_{\mathrm{F}},$$

*where $\sigma_i(\mathbf{M})$ is the $i$th largest singular value of $\mathbf{M}$.*

*Proof.* By the Cauchy-Schwartz inequality,

$$\sum_{i=r+1}^{r+k} \sigma_i(\mathbf{M}) = \sum_{i=r+1}^{r+k} |\sigma_i(\mathbf{M})| \le \left(\sum_{i=r+1}^{r+k} \sigma_i^2(\mathbf{M})\right)^{1/2} \cdot \|\mathbf{1}_k\|_2 = \sqrt{k} \cdot \left(\sum_{i=r+1}^{r+k} \sigma_i^2(\mathbf{M})\right)^{1/2}.$$

Note that $\sigma_{r+1}(\mathbf{M}), \dots, \sigma_{r+k}(\mathbf{M})$ are the $k$ smallest among the $r+k$ largest singular values. Hence,

$$\sum_{i=r+1}^{r+k} \sigma_i^2(\mathbf{M}) \le \frac{k}{r+k} \sum_{i=1}^{r+k} \sigma_i^2(\mathbf{M}) \le \frac{k}{r+k} \sum_{i=1}^{\min\{m,n\}} \sigma_i^2(\mathbf{M}) = \frac{k}{r+k} \|\mathbf{M}\|_{\mathrm{F}}^2.$$

Combining the two inequalities, the desired result follows. □

**Corollary 1.** *For any real $m \times n$ matrix $\mathbf{M}$ and $k \le \min\{m, n\}$, $\sigma_k(\mathbf{M}) \le k^{-1/2} \cdot \|\mathbf{M}\|_{\mathrm{F}}$.*

*Proof.* It follows immediately from Lemma 6. □

**Lemma 7.** *Let $a_1, \dots, a_n$ and $b_1, \dots, b_n$ be $2n$ real numbers and let $p$ and $q$ be two numbers such that $1/p + 1/q = 1$ and $p > 1$. We have*

$$\left|\sum_{i=1}^{n} a_i b_i\right| \le \left(\sum_{i=1}^{n} |a_i|^p\right)^{1/p} \cdot \left(\sum_{i=1}^{n} |b_i|^q\right)^{1/q}.$$

**Lemma 8.** *For any $\mathbf{A}, \mathbf{B} \in \mathbb{R}^{n \times k}$,*

$$\left|\langle \mathbf{A}, \mathbf{B} \rangle\right| \triangleq \left|\mathrm{Tr}(\mathbf{A}^\top \mathbf{B})\right| \le \|\mathbf{A}\|_{\mathrm{F}} \|\mathbf{B}\|_{\mathrm{F}}.$$

*Proof.* The inequality follows from Lemma 7 for $p = q = 2$, treating $\mathbf{A}$ and $\mathbf{B}$ as vectors. □

**Lemma 9.** *For any two real matrices $\mathbf{A}$ and $\mathbf{B}$ of appropriate dimensions,*

$$\|\mathbf{AB}\|_{\mathrm{F}} \le \min\{\|\mathbf{A}\|_2 \|\mathbf{B}\|_{\mathrm{F}}, \ \|\mathbf{A}\|_{\mathrm{F}} \|\mathbf{B}\|_2\}.$$

*Proof.* Let $\mathbf{b}_i$ denote the $i$th column of $\mathbf{B}$. Then,

$$\|\mathbf{AB}\|_\mathrm{F}^2 = \sum_i \|\mathbf{Ab}_i\|_2^2 \le \sum_i \|\mathbf{A}\|_2^2 \|\mathbf{b}_i\|_2^2 = \|\mathbf{A}\|_2^2 \sum_i \|\mathbf{b}_i\|_2^2 = \|\mathbf{A}\|_2^2 \|\mathbf{B}\|_\mathrm{F}^2.$$

Similarly, using the previous inequality,

$$\|\mathbf{AB}\|_\mathrm{F}^2 = \|\mathbf{B}^\top \mathbf{A}^\top\|_\mathrm{F}^2 \le \|\mathbf{B}^\top\|_2^2 \|\mathbf{A}^\top\|_\mathrm{F}^2 = \|\mathbf{B}\|_2^2 \|\mathbf{A}\|_\mathrm{F}^2.$$

Combining the two upper bounds, the desired result follows. $\square$

**Corollary 2.** *Let $\mathbf{M}_r$ denote the best rank-$r$ approximation of $\mathbf{M}$, obtained by the truncated singular value decomposition of $\mathbf{M}$. Then, for any $d > r$, $\sigma_d \le \|\mathbf{M} - \mathbf{M}_r\|_\mathrm{F}/\sqrt{d-r}$.*

*Proof.* By definition

$$\|\mathbf{M} - \mathbf{M}_r\|_\mathrm{F}^2 = \sum_{i=r+1}^{n} \sigma_i^2 \ge \sum_{i=r+1}^{d} \sigma_i^2 \ge (d-r) \cdot \sigma_d^2,$$

from which the desired result follows. $\square$

**Lemma 10.** *For any real $m \times n$ matrix $\mathbf{A}$, and any $k \le \min\{m,\, n\}$,*

$$\max_{\substack{\mathbf{Y} \in \mathbb{R}^{n \times k} \\ \mathbf{Y}^\top \mathbf{Y} = \mathbf{I}_k}} \|\mathbf{AY}\|_\mathrm{F} = \left( \sum_{i=1}^{k} \sigma_i^2(\mathbf{A}) \right)^{1/2}.$$

*Equality is realized for $\mathbf{Y}$ coinciding with the $k$ leading right singular vectors of $\mathbf{A}$.*

*Proof.* Let $\mathbf{U\Sigma V}^\top$ be the singular value decomposition of $\mathbf{A}$; $\mathbf{U}$ and $\mathbf{V}$ are $m \times m$ and $n \times n$ unitary matrices respectively, while $\Sigma$ is a diagonal matrix with $\Sigma_{jj} = \sigma_j$, the $j$th largest singular value of $\mathbf{A}$, $j = 1, \ldots, d$, where $d \triangleq \min\{m, n\}$. Due to the invariance of the Frobenius norm under unitary multiplication,

$$\|\mathbf{AY}\|_\mathrm{F}^2 = \|\mathbf{U\Sigma V}^\top \mathbf{Y}\|_\mathrm{F}^2 = \|\mathbf{\Sigma V}^\top \mathbf{Y}\|_\mathrm{F}^2. \tag{32}$$

Continuing from (32),

$$\|\mathbf{\Sigma V}^\top \mathbf{Y}\|_\mathrm{F}^2 = \mathrm{TR}\left( \mathbf{Y}^\top \mathbf{V \Sigma}^2 \mathbf{V}^\top \mathbf{Y} \right) = \sum_{i=1}^{k} \mathbf{y}_i^\top \left( \sum_{j=1}^{d} \sigma_j^2 \cdot \mathbf{v}_j \mathbf{v}_j^\top \right) \mathbf{y}_i = \sum_{j=1}^{d} \sigma_j^2 \cdot \sum_{i=1}^{k} \left( \mathbf{v}_j^\top \mathbf{y}_i \right)^2.$$

Let $z_j \triangleq \sum_{i=1}^{k} \left( \mathbf{v}_j^\top \mathbf{y}_i \right)^2$, $j = 1, \ldots, d$. Note that each individual $z_j$ satisfies

$$0 \le z_j \triangleq \sum_{i=1}^{k} \left( \mathbf{v}_j^\top \mathbf{y}_i \right)^2 \le \|\mathbf{v}_j\|^2 = 1,$$

where the last inequality follows from the fact that the columns of $\mathbf{Y}$ are orthonormal. Further,

$$\sum_{j=1}^{d} z_j = \sum_{j=1}^{d} \sum_{i=1}^{k} \left( \mathbf{v}_j^\top \mathbf{y}_i \right)^2 = \sum_{i=1}^{k} \sum_{j=1}^{d} \left( \mathbf{v}_j^\top \mathbf{y}_i \right)^2 = \sum_{i=1}^{k} \|\mathbf{y}_i\|^2 = k.$$

Combining the above, we conclude that

$$\|\mathbf{AY}\|_\mathrm{F}^2 = \sum_{j=1}^{d} \sigma_j^2 \cdot z_j \le \sigma_1^2 + \ldots + \sigma_k^2. \tag{33}$$

Finally, it is straightforward to verify that if $\mathbf{y}_i = \mathbf{v}_i$, $i = 1, \ldots, k$, then (33) holds with equality. $\square$

**Corollary 3.** *For any real $m \times m$ PSD matrix $\mathbf{A}$, and $k \times m$ matrix $\mathbf{X}$ with $k \leq m$ orthonormal columns,*

$$\mathrm{TR}\big(\mathbf{X}^\top \mathbf{A}\mathbf{X}\big) = \sum_{i=1}^{k} \lambda_i(\mathbf{A})$$

*where $\lambda_i(\mathbf{A})$ is the ith largest eigenvalue of $\mathbf{A}$. Equality is achieved for $\mathbf{X}$ coinciding with the $k$ leading eigenvectors of $\mathbf{A}$.*

*Proof.* Let $\mathbf{A} = \mathbf{V}\mathbf{V}^\top$ be a factorization of the PSD matrix $\mathbf{A}$. Then, $\mathrm{TR}\big(\mathbf{X}^\top \mathbf{A}\mathbf{X}\big) = \mathrm{TR}\big(\mathbf{X}^\top \mathbf{V}\mathbf{V}^\top \mathbf{X}\big) = \|\mathbf{V}^\top \mathbf{X}\|_{\mathrm{F}}^2$. The desired result follows by Lemma 10 and the fact that $\lambda_i(\mathbf{A}) = \sigma_i^2(\mathbf{V})$, $i = 1, \ldots, m$. $\qquad\square$

## C  Net of the $\ell_2$-unit sphere

In this section, we provide a simple probabilistic construction of an $\epsilon$-net of the $\ell_2$-unit sphere $\mathbb{S}_2^{d-1}$, *i.e.*, the set of points $\mathbf{x}$ such that $\|\mathbf{x}\|_2 = 1$.

**Lemma 11** ([34], Lemma 5.2). *For any $\epsilon > 0$, there exists an $\epsilon$-net $\mathcal{N}_\epsilon$ of the unit Euclidean sphere $\mathbb{S}_2^{d-1}$ equipped with the Euclidean metric, such that*

$$m_\epsilon \triangleq |\mathcal{N}_\epsilon| \leq (1 + 2/\epsilon)^d.$$

*Proof.* Let $\mathcal{N}_\epsilon$ be a maximal $\epsilon$-separated subset of $\mathbb{S}_2^{d-1}$. In other words, $d(x, y) \geq \epsilon$ for all $x, y \in \mathcal{N}_\epsilon$, $x \neq y$, and no set containing $\mathcal{N}_\epsilon$ has this property.

Such a set can be constructed iteratively: select an arbitrary point on the sphere and at each subsequent step select a point that is at distance at least $\epsilon$ from all previously selected points. By the compactness of the sphere, the iterative construction process will terminate after a finite number of steps, and the resulting set will satisfy the above properties.

The maximality property, implies that $\mathcal{N}_\epsilon$ is an $\epsilon$-net of $\mathbb{S}_2^{d-1}$. If this was not the case, then there would exist an $x \in \mathbb{S}_2^{d-1}$ such that $d(x, y) > \epsilon$, $\forall y \in \mathcal{N}_\epsilon$. The $\mathcal{N}_\epsilon \cup \{x\}$ would an $\epsilon$-separated set, that contains $\mathcal{N}_\epsilon$, contradicting the maximality of the latter.

By the separation property, we infer that the balls of radius $\epsilon/2$ centered at the points of $\mathcal{N}_\epsilon$ are disjoint. This follows from the triangle inequality. Further, all such balls lie in the ball $(1 + \epsilon/2)\,\mathbb{B}_2^d$, where $\mathbb{B}_2^d$ denotes the unit Euclidean ball centered at the origin. Comparing the volumes, we have

$$\mathrm{Vol}\big(\tfrac{\epsilon}{2}\mathbb{B}_2^d\big) \cdot |\mathcal{N}_\epsilon| \leq \mathrm{Vol}\big((1 + \tfrac{\epsilon}{2})\mathbb{B}_2^d\big).$$

Taking into account that $\mathrm{Vol}\big(r \cdot \mathbb{B}_2^d\big) = r^d \cdot \mathrm{Vol}\big(\mathbb{B}_2^d\big)$,

$$|\mathcal{N}_\epsilon| \leq \big(1 + \tfrac{\epsilon}{2}\big)^d / \big(\tfrac{\epsilon}{2}\big)^d = \big(1 + \tfrac{2}{\epsilon}\big)^d,$$

which is the desired result. $\qquad\square$

Lemma 11 regards the unit Euclidean sphere. However, the sequence of arguments used in the proof essentially hold for the case of the unit ball $\mathbb{B}_2^d$, *i.e.*, there exists an $\epsilon$-net of $\mathbb{B}_2^d$, with at most $(1 + 2/\epsilon)^d$ points.

**Constructing an $\epsilon$-net of the unit sphere.** There are many constructions for $\epsilon$-nets on the sphere, both deterministic and randomized. In the following we review a simple randomized construction, initially studied by Wyner [35] in the asymptotic $d \to \infty$ regime.

By Lemma 11, there exists an $\epsilon$-net $\mathcal{N}_\epsilon$ of $\mathbb{S}_2^{d-1}$ Consider the balls of radii $\epsilon$ centered at the points of $\mathcal{N}_\epsilon$. The balls cover all points of $\mathbb{S}_2^{d-1}$; if there existed a point $x$ on $\mathbb{S}_2^{d-1}$ not included in any ball, it would imply that this point is at distance at least $\epsilon$ from all points of $\mathcal{N}_\epsilon$ contradicting the fact that $\mathcal{N}_\epsilon$ is a $\epsilon$-net.

The intersection of each of the previous balls with $\mathbb{S}_2^{d-1}$ is a spherical cap, and hence, according to the above, the collection of sperical caps covers $\mathbb{S}_2^{d-1}$. (Note that the spherical caps, as well as the balls, overlap.)

Consider a set $\mathcal{Q}$, containing at least one point from each spherical cap. Then, $\mathcal{Q}$ is a $2\epsilon$-net of $\mathbb{S}_2^{d-1}$. To verify that, note the following. Consider a point $x \in \mathbb{S}_2^{d-1}$. By construction, $\mathcal{N}_\epsilon$ contains a point $y$ such that $d(x, y) \le \epsilon$. Consider the spherical cap centered at $y$. By definition, $\mathcal{Q}$ contains a point $\widetilde{y}$ in that spherical cap, and hence $d(y, \widetilde{y}) \le \epsilon$. By triangle inequality, it follows that $d(x, \widetilde{y}) \le 2\epsilon$. Since the point $x$ is arbitrary, we conclude that $\mathcal{Q}$ is a $2\epsilon$-net.

We draw points randomly and independently, uniformly distributed on $\mathbb{S}_2^{d-1}$. This can be accomplished, for instance, by randomly and independently generating vectors in $\mathbb{R}^d$ distributed according to the a the multivariate normal distribution $N(\mathbf{0}, \mathbf{I})$ and normalizing their length to 1. A randomly selected point lies in a specific spherical cap with probability $p \ge 1/m_\epsilon$. By a standard probability arguments (Coupon collector's problem), $O(m_\epsilon \ln(m_\epsilon/\delta))$ points uniformly distributed over $\mathbb{S}_2^{d-1}$ suffice for at least one random point to lie in each sphere cap with probability at least $1 - \delta$. Substituting the value of $m_\epsilon$ from Lemma 11, we find that $O\left(d\epsilon^{-d} \cdot \ln \frac{1}{\epsilon\delta}\right)$ suffice to form a $2\epsilon$-net, with probability at least $1 - \delta$. Note that $\delta$ can be chosen to scale with the dimension $n$ of the problem.

**Lemma 12.** *A set of* $O\left(d(\epsilon/2)^{-d} \cdot \ln \frac{2}{\epsilon \cdot \delta}\right)$ *randomly and independently drawn points uniformly distributed on* $\mathbb{S}_2^{d-1}$ *suffices to construct an* $\epsilon$*-net of* $\mathbb{S}_2^{d-1}$ *with probability at least* $1 - \delta$.

## D    Additional Experimental Results

| Component 1 | Component 2 | Component 3 | Component 4 | Component 5 |
|---|---|---|---|---|
| american | coach | add | ago | billion |
| attack | game | cup | called | business |
| campaign | games | food | com | companies |
| country | guy | hour | family | company |
| government | hit | large | help | cost |
| group | left | makes | high | deal |
| leader | night | minutes | home | industry |
| official | play | oil | look | market |
| political | player | pepper | need | million |
| president | point | serving | part | money |
| zzz_al_gore | run | small | problem | number |
| zzz_bush | season | sugar | program | percent |
| zzz_george_bush | team | tablespoon | right | plan |
| zzz_u_s | win | teaspoon | school | stock |
| zzz_united_states | won | water | show | system |

Table 3: ONMF with $r = 5$ orthogonal components ($102 \cdot 10^3$–dimensional vectors) on the words-by-document matrix of the NY Times bag-of-words dataset [33]. The table depicts the words corresponding to the 15 largest entries of each component. The 5 retrieved components are extremely sparse: $90\%$ of their mass is concentrated in 134, 35, 65, 269 and 59 entries, respectively.

**Large-scale text analysis: clustering words.**    We evaluate the performance of ONMFS as a clustering algorithm on the NY Times bag-of-words dataset [33]. The dataset is represented by a $102K \times 300K$ words-by-articles matrix. Given an approximate ONMF of that matrix, the $102K \times k$ nonnegative, orthogonal factor $\mathbf{W}$ induces an assignment of words to $r$ clusters, which in this case can be interpreted as *topics*. That is, each column of $\mathbf{W}$ suggests a topic, defined by the words corresponding to its nonzero entries.

We run ONMFS with target dimension $k = 5$ topics, and accuracy parameter $r = 5$, while we configure it to stop if no progress is observed after $T = 300$ consecutive candidate solutions. Table 3 lists the words corresponding to the 15 largest entries of each orthogonal component (column of $\mathbf{W}$). Arguably, each component can be interpreted as a distinct topic, illustrating the potential of ONMF in text analysis. Further, we note that although the components of $\mathbf{W}$ are not explicitly restricted to

be sparse, they tend to be: $90\%$ of the $\ell_2$ mass of each component is concentrated in approximately 100-200 entries (words) out of the roughly 102K present in the dataset.

## Appendix References

[34] Roman Vershynin. Introduction to the non-asymptotic analysis of random matrices. *arXiv preprint arXiv:1011.3027*, 2010.

[35] Aaron D Wyner. Random packings and coverings of the unit n-sphere. *Bell System Technical Journal*, 46(9):2111–2118, 1967.