[Reviews · NeurIPS 2015]

Submitted by Assigned_Reviewer_1

The paper proposes a subspace exploration algorithm for the problems of orthogonal NMF and nonnegative PCA (which are tightly related). According to the authors, this is the first algorithm that comes with performance guarantees that are characterized by a tradeoff between the approximation error and the computational complexity. Although the complexity scales exponentially with some user controlled parameters (order of low rank approximation, and target rank), simulations indicate good performance in practice.

Quality: The paper is technically sound, with proofs about the main theorems, discussion about their shortcomings (exponential complexity), and simulations to establish state of the art results in practice.

Clarity: The paper is mostly clearly written. I think one clarifications that is important but is missing is the following: Algorithm 3 LocalOptW runs with complexity O(2^k*k*m). Naively, for Algorithm 1, algorithm 3 has to be run for every choice of C. But the number of choices for C is also exponentially large (the authors should state explicitly how large it is). This would make the whole algorithm even more computationally hard. However the authors argue (footnote 2) that when the two algorithms are combined (as in the application shown in this paper), Alg 3 can be simplified into a O(k*m) procedure. While intuitive, the authors should make explicitly clear how this is done, and perhaps re-write algs 1 and 3 to demonstrate that.

Originality: To the best of my knowledge the approach is original.

Significance: The paper establishes both theoretical bounds (for the first time), and state of the art results through simulations. I'd like to see a comparison (or a discussion) of the various different methods, also with respect to the run-time of the various algorithms. The complexity bound O((1/\epsilon)^(1/\epsilon)) can grow rapidly, so it'd be nice to see that the gained performance does not come at a terrible computational cost.

Since ONMF and NNPCA are largely equivalent, why does the proposed method perform better than EM for NNPCA (table 1) but worse than EM for ONMF (table 2) for the same dataset MFEAT PIX? Is there something that I miss here?

Typo: last paragraph of the results section, Table 1 should be Table 2 (I think)
Summary: The paper establishes for the first time performance bounds for the problems of ONMF and NNPCA though an algorithm that has exponential complexity. Some additional clarifications are needed before possible publication.

Submitted by Assigned_Reviewer_2

The paper proposes an algorithm for solving nonnegative matrix factorization subject to ortogonality of factors. The algorithm comes with theoretical guarantee of the approximation and with bounds on the polynomial computation time. The motivation for imposing orthogonality on NMF (separation of components) is well-emphasized and theoretical findings are illustrated on numerical experiments (synthetic and real-data). The paper contains 3 Algorithms: (A1) LowRankNNPCA, (A2) ONMFS (A3) LocalOptW. (A2) uses (A1) to solve the problem in Eq. (1). (A1) can be used independently for solving problem Eq. (3). (A1) calls (A3) as a subroutine multiple times, after having denoised the input matrix using a rank-r approximation. (A3) performs a greedy search over all sign configurations (orthants) to optimize the sum of squares objective over the non-convex domain W_k. The cost is exponential in k but k is supposed to be small.

I overall enjoyed reading the paper and think its results are interesting.

There is a key element in A1 which is the weakness of this work. The search over the epsilon-net N_eps^k(S_2^r) can be prohibitively expensive. Authors admit that the difficulty of NMF is hidden here. The difficulty of this step also limits reproducibility and extensions of the approach.

Minor remarks.

The notation T_{SVD} (computation time of SVD) is introduced without definition. There is an ambiguity: I guess it means complexity of computing (r+1) SVD, not full SVD, which can be computed faster than a full SVD when r is small. This is a point in favor of the approach which can be more clear there.

The notation for this N_.. also needs to be defined earlier in the text. the caligraphed N is usually used for normal distribution.
Summary: The paper studies NMF with orthogonality constraints. Algorithms come with theoretical guarantees and numerical experiments. Overall well-written

Submitted by Assigned_Reviewer_3

The paper proposes a new algorithm for NNPCA and ONMF, with some theoretical warranty on the given approximation. It is the first that such warranty are provided in this context. The theoretical results are nice.

The first part of the paper is very well written, until Sec. 2.2. The problem is nicely presented as well as the state of the art. The beginning of Sec. 2.2 is difficult to read. The authors tried to give a "simple" overview of their algorithm to presents the theoretical results, but this overview is not very clear : some important notations appear only 3 pages later (I think in particular on the \mathcal{N}_{\epsilon/2}...).

The theoretical results are clear, but we only understand at this stage that it relies on a first SVD approximation of the data matrix M, the main loop of Alg. 1 remains unintuitive.

Sec. 3 goes back on the algorithm. Unfortunately, the algorithm remains very difficult to understand, particularly without the supplementary materials. I did not understand the construction of the epsilon-net, which seems very important to obtain an efficient algorithm.

The experimental section is clear, but suffers from important drawbacks. The only showed results are the explained variance on various dataset, compared to various approached. As ONMF seems adequat for classification, some results on a classification performances would be very welcome ! Moreover, some computational time comparison would also be nice.

finally, I think it would be interested to mention the "Projective NMF" ([a]), which aims to give a "simple" approximation of the ONMF (without warranty of the orthogonality of W), but which is very easy to optimize.

[a] Yuan, Z. and Oja, E., Projective nonnegative matrix factorization for image compression and feature extraction. In: Image Analysis Springer, Berlin, Germany, pp. 333-342, 2005.
Summary: An interesting paper with a new algorithm for NNPCA and ONMF, but sometimes very difficult to read, and with some lack in the experimental section.

Author Feedback
Author rebuttal: We are grateful that the reviewers enjoyed reading our paper and appreciate their positive feedback.
We now address the remarks of Reviewers 1, 2 and 3.

REVIEWER 1

* [...] Algorithm 3 LocalOptW runs with complexity O(2^k*k*m).[...] algorithm 3 has to be run for every choice of C. But the number of choices for C is also exponentially large [...] the authors argue [...] when the two algorithms are combined [...] Alg 3 can be simplified into a O(k*m) procedure. While intuitive, the authors should make explicitly clear how this is done [...].

Indeed, Alg 3 is called as a subroutine by Alg 1 for each choice of C. That is explicitly reflected in the complexity of Alg 1 (Thm 1); the latter captures the size of the net. (Similarly for Alg 2 in Thm 2). The footnote points out that an additional factor of 2^k in the complexity of Alg 3 can be saved when Alg 3 is used as a subroutine of Alg 1 instead of a standalone procedure. That is a minor remark since it does not affect the asymptotic complexity expression for Alg 1. We will make this more clear.

* I'd like to see a comparison [...] of the various different methods, also with respect to the run-time [...]. The complexity bound [...] can grow rapidly, so it'd be nice to see that the gained performance does not come at a terrible computational cost.

Indeed, the complexity scales unfavorably in epsilon and is prohibitive from a practical point of view. In theory, it is interesting because it yields a novel guarantee for the ONMF problem. In practice, for small epsilon our algorithm is slower than the other methods. In our experiments, we terminate our algorithm after a few samples have been examined, and still get competitive results. All methods run for a few minutes. Also, note that our algorithm allows for a certain degree of parallelization, not exploited here.

* Since ONMF and NNPCA are largely equivalent, why does the proposed method perform better than EM for NNPCA [...] but worse than EM for ONMF [...] for the same dataset [...]? Is there something that I miss here?

This point is indeed confusing. These are two different algorithms. The EM algorithm for NNPCA is from [26], while the one for ONMF from [11]. They both rely on an EM-based approach. Given that our algorithm outperformed the other approaches in the NNPCA objective, we did not use them for the ONMF experiments.

REVIEWER 2

* There is a key element in A1 which I wish to understand better. The search over the epsilon-net N_eps^k(S_2^r) can be prohibitively expensive. The notation for this N_.. also needs to be defined earlier in the text.

Indeed, the size of the epsilon-net is the computational bottleneck of our approach. Determining whether it can be improved or showing it cannot is an open problem.
The notation for the epsilon-net is admittedly not ideal! We will move the definition earlier in the text.

* The notation T_{SVD} (computation time of SVD) is introduced without definition. There is an ambiguity: I guess it means complexity of computing (r+1) SVD, [...]. This is a point in favor of the approach which can be more clear there.

That is correct and will be addressed in the final manuscript.

REVIEWER 3

* The beginning of Sec. 2.2 is difficult to read. The authors tried to give a "simple" overview of their algorithm to presents the theoretical results, but this overview is not very clear : some important notations appear only 3 pages later (I think in particular on the \mathcal{N}_{\epsilon/2}...).
The theoretical results are clear, but [...] the main loop of Alg. 1 remains unintuitive.

The reviewer is right. We felt that a more "intuitive" description of Alg 1 would also require a more technical description. Given the available space, we decided to postpone the deeper technical description of the main algorithm until Sec 3 and present the main theoretical results first. We will at least ensure that all notation is properly introduced and try to improve that section.

* I did not understand the construction of the epsilon-net, which seems very important to obtain an efficient algorithm.

To obtain our results, it suffices to consider a simple net obtained by sampling r-dim points i.i.d. from a Gaussian distribution and normalizing their length. The size of the net is indeed the computational bottleneck of our approach.

* The experimental section is clear, but suffers from important drawbacks. [...] As ONMF seems adequate for classification, some results on a classification performances would be very welcome ! Moreover, some computational time comparison would also be nice.

It would indeed be interesting to include such results in the supplemental material. Most importantly, we should have included a discussion on the time comparison among various methods. We will try to address that in the final manuscript.